



# Development of the tangent linear and adjoint models of the global online chemical transport model MPAS-CO$_2$ v7.3

Tao Zheng[1,7], Sha Feng[2], Jeffrey Steward[3], Xiaoxu Tian[4], David Baker[5], and Martin Baxter[6]

[1]Department of Geography and Environmental Studies, Central Michigan University, Mount Pleasant, MI. USA
[2]Atmospheric Sciences and Global Change Division, Pacific Northwest National Laboratory, Richland, WA. USA
[3]Data Assimilation Intelligence for Space Systems, LLC., Boulder, CO. USA
[4]Tomorrow.io, Boston, MA. USA
[5]Cooperative Institute for Research in the Atmosphere, Colorado State University, Fort Collins, CO. USA
[6]Department of Earth and Atmospheric Science, Central Michigan University, Mount Pleasant, MI. 48858
[7]Institute for Great Lakes Research, Central Michigan University, Mount Pleasant, MI. USA

**Correspondence:** Tao Zheng(zheng1t@cmich.edu)

**Abstract.** We describe the development of the tangent linear (TL) and adjoint models of the MPAS-CO$_2$ transport model, which is a global online chemical transport model developed upon the non-hydrostatic Model for Prediction Across Scales-Atmosphere (MPAS-A). The primary goal is to make the model system a valuable research tool for investigating atmospheric carbon transport and inverse modeling. First, we develop the TL code, encompassing all CO$_2$ transport processes within

5 the MPAS-CO$_2$ forward model. Then, we construct the adjoint model using a combined strategy involving re-calculation and storage of the essential meteorological variables needed for CO$_2$ transport. This strategy allows the adjoint model to undertake long-period integration with moderate memory demands. To ensure accuracy, the TL and adjoint models undergo vigorous verifications through a series of standard tests. The adjoint model, through backward-in-time integration, calculates the sensitivity of atmospheric CO$_2$ observations to surface CO$_2$ fluxes and the initial atmospheric CO$_2$ mixing ratio. To

10 demonstrate the utility of the newly-developed adjoint model, we conduct simulations for two types of atmospheric CO$_2$ observations: tower-based *in situ* CO$_2$ mixing ratio and satellite-derived column-averaged ($X_{CO_2}$). A comparison between the sensitivity to surface flux calculated by the MPAS-CO$_2$ adjoint model with its counterpart from Carbon Tracker-Lagrange (CT-L) reveals spatial agreement but notable magnitude differences. These differences, particularly evident for $X_{CO_2}$, likely arise from differences in vertical mixing between the two systems. Moreover, this comparison highlights the substantial loss of

15 information in the atmospheric CO$_2$ observations due to CT-L's simulation length and spatial domain limitations. Furthermore, the adjoint sensitivity analysis demonstrates that the sensitivities to both surface flux and initial CO$_2$ conditions spread out throughout the entire northern hemisphere within a month. MPAS-CO$_2$ forward, TL, and adjoint models stand out for their calculation efficiency and variable-resolution capability, making them competitive in computational cost. In conclusion, the successful development of the MPAS-CO$_2$ TL and adjoint models, and their integration into the MPAS-CO$_2$ system, establish

the possibility of using MPAS's unique features in atmospheric CO$_2$ transport sensitivity studies and in inverse modeling with advanced methods such as variational data assimilation.



## 1 Introduction

Estimating $CO_2$ fluxes through inverse modeling, using atmospheric chemical transport models and atmospheric $CO_2$ measurements, is an important approach for understanding the global carbon budget. Beyond providing seasonal flux estimates that are useful for understanding the magnitude and phase of photosynthesis and respiration, it provides annual mean flux estimates that shed light on the key processes driving the response to climate change. When these annual mean $CO_2$ estimates are adjusted to account for lateral fluxes (e.g., due to rivers, or the transport of crops and wood products), it gives an independent means of validating carbon stock change estimates from the terrestrial biogeochemical models and inventories (Byrne et al., 2023). However, atmospheric transport models, which play a key role in inverse modeling, remain a significant source of uncertainty on both regional and global scales (Hurtt et al., 2022).

Two classes of chemical transport models – online and offline – are commonly used for simulating atmospheric $CO_2$ transport. Offline models, such as TM5 (Krol et al., 2005; Meirink et al., 2006), PCTM (Kawa et al., 2004; Baker et al., 2006) and GEOS-Chem (Kopacz et al., 2009), solve the tracer continuity equation using winds and vertical mixing fields computed from an independent run of a meteorological model or from a meteorological analysis. Online models, such as WRF-Chem (Grell et al., 2011), OLAM (Walko and Avissar, 2008; Schuh et al., 2021), and MPAS-$CO_2$ (Skamarock et al., 2012; Zheng et al., 2021), integrate chemistry, transport and meteorology simultaneously. Although offline models typically have lower computational costs, the separation of chemistry/transport from meteorology leads to a loss of information regarding atmospheric processes occurring at time scales shorter than the meteorological model output frequency (Grell et al., 2005). In comparison, online models, owing to their simultaneous integration of meteorology and chemistry, have the potential to improve transport accuracy, particularly for vertical transport of chemistry. Recent advances in computer power and parallelization have greatly reduced the computational cost of online transport models, making them increasingly more accessible and practical for atmospheric $CO_2$ research.

A number of studies have demonstrated that transport model accuracy can be improved by increasing the model's horizontal resolution (Feng et al., 2016; Agusti-Panareda et al., 2019). Because global high-resolution $CO_2$ transport simulations are computationally demanding, limited-area models (regional models) are often used instead (Pillai et al., 2012; Lauvaux et al., 2012; Zheng et al., 2018). However, regional models introduce the lateral boundary condition, posing challenges for $CO_2$ inverse modeling (Zheng et al., 2019; Rayner et al., 2019). MPAS-$CO_2$ (Zheng et al., 2021) addresses this limitation by being an online global transport model based on the compressible non-hydrostatic Model for Prediction Across Scales-Atmosphere (MPAS-A) (Skamarock et al., 2012). Like OLAM (Schuh et al., 2021), MPAS-$CO_2$ uses a global variable-resolution mesh to facilitate local grid refinement for high-resolution simulations in specific regions without incurring prohibitively high computational costs and avoiding the disadvantages of lateral boundary conditions.





The primary objective of this study is to develop the tangent linear (TL) and adjoint (AD) models associated with the global online transport model MPAS-CO$_2$ (Zheng et al., 2021). Adjoint model techniques have been widely used in both meteorological and atmospheric greenhouse gas research (Errico, 1997; Courtier et al., 1994; Giering et al., 2006; Meirink et al., 2008; Henze et al., 2007; Tian and Zou, 2021), and play critical roles in variational data assimilation and sensitivity analyses (Baker et al., 2006; Zheng et al., 2018; Tian and Zou, 2020).

The subsequent sections of this paper provide an overview of the MPAS-CO$_2$ forward model developed in Zheng et al. (2021) (Section 2), and the development and verification of the TL and AD models based on the forward model (Sections 3 & 4). The utility of the newly developed AD model is demonstrated with adjoint sensitivity analyses in Section 5. Finally, a summary and conclusion are given in Section 6.

## 2 MPAS-CO$_2$ forward model

Zheng et al. (2021) documented the development of MPAS-CO$_2$, verifying its mass conservation and assessing its accuracy. Hereafter, we refer to MPAS-CO$_2$ as the forward model, whose TL and AD model counterparts we develop in the present paper. A brief description of the forward model is provided here; see Zheng et al. (2021) for comprehensive details. The forward model characterizes CO$_2$ transport through the continuity equation:

$$\frac{\partial(\tilde{\rho}q_{co_2})}{\partial t} = -(\nabla \cdot \tilde{\rho}q_{co_2}\mathbf{V})_\zeta + F_{bl} + F_{cu} \tag{1}$$

where $q_{co_2}$ is CO$_2$ dry air mixing ratio, $\tilde{\rho} = \rho_d/(\partial\zeta/\partial z)$, $\rho_d$ is dry air density, $\zeta$ is the vertical coordinate, $z$ is geometric height, $t$ is time, and $\mathbf{V} = (u, v, w)$ is the velocity vector ($u$, $v$, and $w$ are the zonal, meridional, and vertical wind components, respectively). The meteorological variables, such as wind velocity and dry air density, are updated simultaneously with CO$_2$ by the model's dynamical core and physics parameterizations. The left-hand side (LHS) of Eq. (1) is the total CO$_2$ time tendency ($\partial(\tilde{\rho}q_{co_2})/\partial t$), and the first, second, and third terms on the right-hand side (RHS) represent the contributions from advection, vertical mixing by the planetary boundary layer (PBL) parameterization, and convective transport, respectively. Advection of CO$_2$ in MPAS-CO$_2$ is handled in the model's dynamical core and can be expressed as Eq. (2), where the first two terms on the RHS represent the horizontal advection, and the third term represents the vertical advection:

$$(\nabla \cdot \tilde{\rho}q_{co_2}\mathbf{V})_\zeta = \Big[\frac{\partial(\tilde{\rho}uq_{co_2})}{\partial x} + \frac{\partial(\tilde{\rho}vq_{co_2})}{\partial y}\Big]_\zeta + \frac{\partial(\tilde{\rho}wq_{co_2})}{\partial \zeta} \tag{2}$$

CO$_2$ vertical mixing in the PBL is implemented based on the YSU PBL scheme (Hong et al., 2006) and can be expressed as:

$$F_{bl} = \frac{\partial q_{co_2}}{\partial t} = \frac{\partial}{\partial z}\Big[K_h\big(\frac{\partial q_{co_2}}{\partial z}\big) - \overline{(w'q'_{co_2})_h}\big(\frac{z}{h}\big)^3\Big] \tag{3}$$





where $z$ is the vertical distance to surface, $h$ is boundary layer top height, and $K_h$ is vertical eddy diffusivity. Convective transport of $CO_2$ is implemented based on the Kain-Fristch convection scheme (Kain, 2004) and it can be expressed as Eq. (4)

85
$$F_{cu} = \frac{\partial q_{co_2}}{\partial t} = \frac{(M_u + M_d)}{\rho A} \frac{\partial q_{co_2}}{\partial z} + \frac{M_{ud}}{M}(q_{co_2}^u - q_{co_2}) + \frac{M_{dd}}{M}(q_{co_2}^d - q_{co_2}) \qquad (4)$$

where $q_{co_2}$, $q_{co_2}^u$, and $q_{co_2}^d$ are the $CO_2$ mixing ratio in the environment, updraft, and downdraft, respectively, $M_u$ and $M_d$ are the updraft and downdraft mass, respectively, $\rho$ is the environment air density, $A$ is the horizontal area of a cell, $M = \rho A \delta z$ is the mass of environmental air in a grid box, and $M_{ud}$ and $M_{dd}$ are the detrainment from the updraft and downdraft, respectively.

## 90  3   Development of the MPAS-$CO_2$ TL model

The $CO_2$ advective transport process described in Eq. (2) is implemented by two two different numerical schemes in the forward model: (1) a monotonic scheme with hyperviscosity ($\beta$) set to 0.25; and (2) a non-monotonic scheme with $\beta = 1.0$ (Skamarock et al., 2012). The monotonicity in the first scheme is achieved by applying a flux limiter in the last step of the third-order Runge-Kutta solver (Wang et al., 2009; Skamarock and Gassmann, 2011). While the second scheme is linear in
$CO_2$, the first scheme is nonlinear due to the application of the flux limiter. Because both the YSU PBL and Kain-Fristch convection schemes are linear in $CO_2$, using the linear advective scheme makes the forward model a linear model in $CO_2$. In this paper, we develop the TL and adjoint models based on the linear version of the MPAS-$CO_2$ forward model, which can be symbolically expressed as:

$$\mathbf{x}_t = \mathcal{M}(\mathbf{x}_0, \mathbf{e}), \qquad (5)$$

where $\mathbf{x}_0$ and $\mathbf{x}_t$ are the $CO_2$ dry air mixing ratio at the initial and forecast time ($t$), respectively. $\mathcal{M}(\ )$ represents the MPAS-$CO_2$ forward model and $\mathbf{e}$ represents a timeseries of $CO_2$ fluxes between times 0 and $t$. While both $\mathbf{x}_0$ and $\mathbf{x}_t$ are 3-dimensional vectors, $\mathbf{e}$ is 2-dimensional, indicating that $CO_2$ flux is applied only to the model's surface cells. Eq.(5) indicates that $CO_2$ mixing ratio at a forecast time ($\mathbf{x}_t$) is determined by the $CO_2$ mixing ratio at an initial time ($\mathbf{x}_0$) and the $CO_2$ flux ($\mathbf{e}$) through the forward model.

The TL and adjoint models are designed to calculate the sensitivity of $\mathbf{X}_t$ with respect to $\mathbf{x}_0$ and $\mathbf{e}$. This is achieved by introducing the TL and adjoint variables of their counterparts in the forward model (Giles and Pierce, 2000). While the introduction of the TL and adjoint variables for the initial $CO_2$ mixing ratio ($\mathbf{x}_0$) is straightforward, it is a bit more complex for the $CO_2$ fluxes ($\mathbf{e}$). This complexity arises from the fact that $CO_2$ flux, at each surface cell of the model, varies with time
throughout the model's entire simulation period. Depending on the underlying biosphere model and emission inventory used, $CO_2$ flux varies at a certain temporal frequency, ranging from hourly to monthly. Although it is possible to introduce TL and





adjoint variables for $CO_2$ flux at the flux's temporal frequency, it is neither practical nor necessary to do so. Instead, a common approach is to introduce flux scaling factors (Henze et al., 2007; Zheng et al., 2018) as follows:

$$\mathbf{e} = \mathbf{S}(\mathbf{k})\tilde{\mathbf{e}}, \tag{6}$$

where $\tilde{\mathbf{e}}$ are time-variant $CO_2$ fluxes, typically from a process model or inventory, and $\mathbf{S}(\mathbf{k})$ is a generic scaling function. Eq. (6) means that at each surface cell, the magnitude of the $CO_2$ flux ($\tilde{\mathbf{e}}$) is adjusted using a flux scaling factor before it is used to modify the cell's $CO_2$ mixing ratio. We implemented Eq. (6) in an emission driver of the forward model in a way that allows the flexibility of choosing the temporal frequency of the flux scaling factor. For instance, for a 24-hour forward model simulation forced by 3-hourly $CO_2$ flux, one can choose to have eight scaling factors at each surface cell (one for each of

the eight 3-hour segments), or just one scaling factor for the entire time period. All the MPAS-$CO_2$ model runs used in the remainder of this paper are conducted using a single scaling factor for each surface cell that is repeated for each flux timestep in the entire simulation period. In this case, the scaling function $\mathbf{S}(\mathbf{k})$ in Eq. (6) is a function of a scaling vector $\mathbf{k}$ that has the same dimension as the model's surface mesh. The introduction of the flux scaling factors turns $CO_2$ flux from active variables to parameters, and the impacts of their variation on $CO_2$ mixing ratio are calculated through their corresponding scaling factors

$\mathbf{k}$. Accordingly, the MPAS-$CO_2$ forward model can be symbolically expressed as

$$\mathbf{x}_t = \mathcal{M}(\mathbf{x}_0, \mathbf{k}) \tag{7}$$

Eq. (7) shows that for a given set of $CO_2$ flux ($\tilde{\mathbf{e}}$), the forecast time $CO_2$ mixing ratio ($\mathbf{x}_t$) is a function of the initial time $CO_2$ mixing ratio ($\mathbf{x_0}$) and the flux scaling factor ($\mathbf{k}$).

The TL counterpart of the MPAS-$CO_2$ forward model represented by Eq. (7) can be symbolically expressed as the first derivative of the forward model:

$$\Delta\mathbf{x}_t = \mathbf{M}(\Delta\mathbf{x}_0, \Delta\mathbf{k}) \tag{8}$$

where $\mathbf{M}(\ )$ represents the MPAS-$CO_2$ TL model, $\Delta\mathbf{x}_0$ and $\Delta\mathbf{x}_t$ are the TL variable of $CO_2$ mixing ratio at the initial and forecast time, respectively, and $\Delta\mathbf{k}$ is the TL variable of the flux scaling factor $\mathbf{k}$. In essence, Eq. (8) shows that the TL model

computes the perturbation in the forecast time $CO_2$ mixing ratio ($\Delta\mathbf{x}_t$), given the perturbation in the flux scaling factor ($\Delta\mathbf{k}$) and/or perturbation in the initial time $CO_2$ mixing ratio ($\Delta\mathbf{x}_0$).

Based on the source code of the forward model, we developed the TL code by differentiating each process relevant to $CO_2$ flux and transport, including advection, vertical mixing by the YSU PBL scheme, convective transport by the Kain-Fritsch

scheme, and the $CO_2$ emission driver that implements Eq. (6). Automatic differentiation tools, such as Tapenade (Hascoet and Pascual, 2013) and Tangent and Adjoint Model Compiler (Giering and Kaminski, 1998), can be used to assist TL and adjoint





code generation. However, the code these tools generate typically contains redundancies and is difficult to read, particularly for the adjoint code. To optimize the computation efficiency and facilitate future code upgrading, we manually developed the TL and adjoint code for MPAS-$CO_2$ with some minor assistance from Tapenade.

After the TL model is completed, a thorough examination of its correctness was undertaken. As indicated in Eq. (8), the TL model can calculate the sensitivity of $x_t$ with respect to both $x_0$ and $k$. The calculation of the sensitivity of $x_t$ with respect to $x_0$ involves the TL code of all the $CO_2$ transport processes, including advection, PBL, and convective transport. In comparison, the calculation of the sensitivity of $x_t$ with respect to the flux scaling factor $k$ involves the TL code of the $CO_2$ emission driver

in addition to the TL code of all the $CO_2$ transport processes. Because the calculation of sensitivity to $k$ includes the TL code of all the processes in the TL model, and because both the transport processes and emission driver are linear, the correctness of the entire MPAS-$CO_2$ TL model can be verified by checking whether the following equation is satisfied (Errico, 1997; Tian and Zou, 2020):

$$\Phi(\alpha) = \frac{\| \mathcal{M}(\mathbf{x_0}, (1+\alpha)\mathbf{k}) - \mathcal{M}(\mathbf{x_0}, \mathbf{k}) \|}{\| \mathbf{M}(0, \alpha\mathbf{k}) \|} = 1, \qquad (9)$$

where $\mathbf{M}()$ is the TL model, $\mathcal{M}()$ is the forward model and $\alpha$ is a scalar. The second item in the numerator of Eq. (9), $\mathcal{M}(\mathbf{x_0}, \mathbf{k})$, is a forward model run. The first item in the numerator, $\mathcal{M}(\mathbf{x_0}, (1+\alpha)\mathbf{k})$ is an identical forward model run except that its flux scaling factor at each surface cell is adjusted by multiplying $1+\alpha$. In the denominator, $\mathbf{M}(0, \alpha\mathbf{k}))$ is a TL model run with its perturbation in initial time $CO_2$ mixing ratio set to zero ($\Delta\mathbf{x}_0 = 0$) and perturbation in flux scaling factor $\Delta\mathbf{k} = \alpha\mathbf{k}$, which is the difference in the flux scaling factors between the two forward model runs.

If the TL model is correctly coded with regard to the forward model, Eq. (9) should be satisfied to the extent of machine accuracy until $\alpha$ is too small that the result is affected by round-off errors and drifts away from unity. To verify using Eq. (9), we ran a series of simulations using the forward model and newly developed tangent linear model with the scale factor $\alpha$ ranging from $1.0 \times 10^3$ to $1.0 \times 10^{-4}$ (Table 1). All the simulations start at 2018-10-01 00:00 UTC, run for 1 month, and end

at 2018-11-01 00:00 UTC. The meteorological initial condition is from the ERA5 reanalysis (Hoffmann et al., 2019) and the $CO_2$ initial condition ($\mathbf{x}_0$) is from the Carbon Tracker (Jacobson et al., 2020) v2022 (CT2022) posterior $CO_2$ mole fraction at this time. Three-hourly $CO_2$ fluxes for the biogenic, fire, fossil fuel, and oceanic components from the CT2022 posterior are applied throughout the 1-month simulation period for each model run. Flux scaling factors of $\mathbf{k} = \mathbf{1}$ were used in all our simulations here, with $\mathbf{1}$ being a vector the same length as $\mathbf{k}$ with ones in every element. The model simulations are conducted

using the global variable-resolution (VR) mesh shown in Fig. 1. This VR mesh has a total of 15,898 cells, which range from 120 km over most of the land regions to 480 km over oceans. Table 1 shows that the magnitudes of both the numerator and the denominator in (9) decrease as $\alpha$ decreases. Moreover, the table also shows that the ratio remains close to unity until $\alpha$ decreases to $1.0 \times 10^{-1}$, beyond which round-off errors lead to a deviation from unity. These results confirm that the MPAS-$CO_2$ TL model has been correctly developed with regard to the forward model. In the next section, we proceed to develop the





MPAS-CO$_2$ adjoint model.

## 4 Development of the MPAS-CO$_2$ adjoint model

An adjoint model is an essential component of a variational data assimilation system and is very useful for adjoint sensitivity analysis. Symbolically, the MPAS-CO$_2$ adjoint model can be expressed as:

$$(\Delta\hat{\mathbf{x}}_0, \Delta\hat{\mathbf{k}}) = \mathbf{M}^T(\Delta\hat{\mathbf{x}}_t), \tag{10}$$

where $\mathbf{M}^T(\ )$ is the MPAS-CO$_2$ adjoint model, $\Delta\hat{\mathbf{k}}$ is the adjoint variable of the flux scaling factor, and $\Delta\hat{\mathbf{x}}_0$ and $\Delta\hat{\mathbf{x}}_t$ are the adjoint variables of CO$_2$ mixing ratio at the initial and forecast time, respectively. Eq. (10) demonstrates that starting with $\Delta\hat{\mathbf{x}}_t$ at the forecast time, the MPAS-CO$_2$ adjoint model runs backward in time to the initial time, resulting in the adjoint variable of CO$_2$ mixing ratio at the initial time ($\Delta\hat{\mathbf{x}}_0$), and the adjoint variable of the flux scaling factor ($\Delta\hat{\mathbf{k}}$).

Similar to its TL model counterpart, the development of the MPAS-CO$_2$ adjoint model was carried out through manual implementation to avoid redundancy and optimize computational efficiency. However, unlike the forward and TL models, the adjoint model faces the challenge of accessing meteorological fields at every time step during its model integration. This challenge arises due to the fact the adjoint model runs backward in time, making the meteorological fields unavailable. One
approach to this problem is saving meteorological fields in memory during the adjoint model's forward sweep, enabling accessing during the subsequent backward sweep (Guerrette and Henze, 2015; Zheng et al., 2018). However, since the MPAS-CO$_2$ adjoint model is intended for long simulations, this approach becomes impractical due to the excessive memory it demands. As an alternative strategy, we adopt an approach that combines both recalculation and storage of the meteorological fields. This strategy effectively divides a long simulation into segments, and the forward and backward sweeps are carried out sequentially
for each segment, requiring internal memory only large enough to accommodate one segment's worth of meteorological fields. This internal manipulation is handled seamlessly by the adjoint model, enabling it to run as long as needed without overburdening memory resources.

The correctness of the newly developed MPAS-CO$_2$ adjoint model can be verified using the following equation (Tian and
200 Zou, 2020):

$$\langle \Delta\mathbf{x}, \mathbf{M}(0, \Delta\mathbf{k}) \rangle = \langle \mathbf{M}^T(\Delta\mathbf{x}), \Delta\mathbf{k} \rangle, \tag{11}$$

where $\langle\ \rangle$ represents the inner product operator, $\Delta\mathbf{x}$ is a perturbation of CO$_2$ mixing ratio and $\Delta\mathbf{k}$ is a perturbation of CO$_2$ flux scaling factor. If the adjoint model is correctly coded with respect to the TL model, Eq. (11) should be satisfied for any choice of $\Delta\mathbf{x}$ and $\Delta\mathbf{k}$. $\mathbf{M}(0, \Delta\mathbf{k})$ on the LHS of the equation is the perturbation in forecast CO$_2$ mixing ratio resulting from





a TL model run whose perturbation in initial $CO_2$ mixing ratio is set to zero and perturbation to flux scaling factor is set to $\Delta\mathbf{k}$. The first item of the RHS, $\mathbf{M}^T(\Delta\mathbf{x})$, represents the adjoint variable of flux scaling factor output from the adjoint model integration from forecast time backward to the initial time. The TL and adjoint model runs on the two sides of Eq. (11) have the same simulation time period, but the latter runs backward in time.

We conducted two sets of experiments using the TL and adjoint models following Eq. (11) to verify the correctness of the newly developed adjoint model. In the first set of experiments, we set $\Delta\mathbf{k} = 10^{-1}\mathbf{1}$, and $\Delta\mathbf{x} = \mathbf{M}(0, \Delta\mathbf{k})$. The experiments were carried out in two steps: First, the TL model was integrated 7 days from the initial time (2018-10-01 00:00 UTC) to the end time (2018-10-08 00:00 UTC), with $\Delta\mathbf{k} = 10^{-1}\mathbf{1}$, resulting in $\mathbf{M}(0, \Delta\mathbf{k})$ which is the perturbation in forecast time $CO_2$ mixing ratio; Second, the adjoint model is initialized at 2018-10-08 00:00 UTC with its adjoint variable for $CO_2$ mixing

set to $\mathbf{M}(0, \Delta\mathbf{k})$. The adjoint model is then integrated backward in time for 7 days to 2018-10-01 00:00 UTC, resulting in $\mathbf{M}^T(\Delta\mathbf{x})$. The LHS and RHS of Eq. (11) are then calculated using the above results (Table 2). The table shows that the agreement between the LHS and RHS of Eq. (11) is about $-5.15 \times 10^{-15}$. This experiment is repeated with the same configuration but the simulation length is increased to 31 days, ending at 2018-11-01 00:00 UTC. As expected, the magnitude of both the LHS and RHS increased, and they agree to about $-2.55 \times 10^{-16}$. In the second set of the experiment, $\Delta\mathbf{k} = 10^{-1}\mathbf{1}$ (same as

the first set of experiments), but $\Delta\mathbf{x} = \mathcal{M}_{14d}(\mathbf{x}_0, \mathbf{k})$, which is the $CO_2$ mixing ratio at the end of 14-day forward model run (2018-10-01 00:00 UTC to 2018-10-15 00:00 UTC). We note that this forward model run uses $\mathbf{x}_0$ from CT2022 posterior $CO_2$ mole fraction, and $\mathbf{k} = \mathbf{1}$, however, Eq. (9) should satisfy for any configurations and simulation period of the forward model. The resulting LHS and RHS of Eq. (11) from the second set of experiments are about 2 orders of magnitude larger than their counterpart of the first experiments. This is caused by the much larger $\Delta\mathbf{x}$ of the second set of experiments. The LHS and RHS

agree to to about $-3.42 \times 10^{-15}$ for the 7-day simulation and about $2.66 \times 10^{-15}$ for the 31-day simulation (Table 2).

   The results shown in Table 2 obtained from the experiments based on Eq. (11) confirm that the MPAS-$CO_2$ adjoint model has been correctly developed with regard to the TL model. As the TL model has already been confirmed correct with respect to the forward model, it follows that both TL and adjoint models are correct with respect to the forward model of MPAS-$CO_2$.

This validation ensures the reliability and integrity of the entire MPAS-$CO_2$ system, including the forward, TL, and adjoint models, as described in this paper and Zheng et al. (2021). It allows MPAS-$CO_2$ to be used as the basis of a variational assimilation system for carbon flux estimation and as a platform for conducting sensitivity analyses in atmospheric carbon research.

   Table 3 presents the computational cost of model simulations using the MPAS-$CO_2$ system. Using the global 120-480 km

VR mesh (Fig. 1; 15898 cells), the 1-month forward model simulation completes in 20 minutes when using 128 processors. Both the TL and adjoint model simulations using the same configuration take approximately 10% longer, indicating that the majority of the computation time is used for integrating the meteorological fields. Furthermore, we conducted another set of 1-month simulations using the models on a global quasi-uniform resolution (UR) mesh of about 120 km, consisting of a total of 40,962 cells. Table 3 demonstrates that the simulations with the VR mesh reduce the computational cost by over 50% for all





three models, primarily due to its substantially smaller number of cells. This reduction in computation cost, while preserving
     the high resolution over areas of interest, should prove advantageous when the models are applied in variational assimilation
     problems, which typically require many iterations of forward and adjoint model runs.

## 5   Adjoint sensitivity analysis

In addition to forming a key component of variational assimilation systems (Baker et al., 2006; Zheng et al., 2018; Tian and
     Zou, 2021), adjoint models are powerful tools for sensitivity analysis (Errico and Vukicevic, 1992; Errico, 1997; Zou et al.,
     1997; Tian and Zou, 2020). Studies focused on carbon flux estimation are often interested in exploring the sensitivity of at-
     mospheric $CO_2$ measurements to surface $CO_2$ fluxes; these maps are commonly referred to as observation influence functions
     or footprints (Cui et al., 2022). The computation of observation footprints using forward models requires a large number of
model runs, making it impractical, except at coarse horizontal resolutions. In contrast, adjoint models can calculate observation
     footprints much more efficiently. For point measurements, such as those from tower data, Lagrangian dispersion models offer
     an efficient alternative for obtaining footprints (Lin et al., 2003; Stohl et al., 2005). For an example of this, see the publicly
     available $CO_2$ observation footprints from Carbon Tracker- Lagrange (CT-L) (Hu et al., 2019), which are generated using the
     Lagrangian particle dispersion model STILT (Lin et al., 2003), driven by meteorology generated by the Weather Research
and Forecast (WRF) model (Skamarock et al., 2008). This approach involves releasing a certain number of particles from the
     observation location/height and tracing their backward transport in time. Note that CT-L is a regional modeling system that
     only provides observation footprints within the latitude range $10^o$-$80^o$ N and longitude $0^o$-$180^o$ W for up to 10 days backward
     in time.

In this section, we perform sensitivity analyses using the MPAS-$CO_2$ adjoint model, which employs backward-in-time inte-
     gration to calculate two quantities: (1) the sensitivity of atmospheric $CO_2$ to the model's initial $CO_2$ mixing ratio, and (2) the
     sensitivity to the surface flux scaling factor. When a uniform time-invariant surface flux is used, the sensitivity to the surface
     flux scaling factor calculated by the MPAS-$CO_2$ adjoint model is the observation footprint. To facilitate comparison with CT-L
     footprints, the MPAS-$CO_2$ adjoint model simulations conducted in this section use uniform time-invariant $CO_2$ surface fluxes
of 1.0 $\mu$mol/(m$^2$ s) for all surface cells, including both land and ocean cells.

         First, we conduct MPAS-$CO_2$ adjoint model simulations for *in situ* $CO_2$ observations at two towers in the United States:
     WKT, located at Moody, Texas ($31.31^o$ N, $97.33^o$ W), and WGC, located at Walnut Grove, California ($38.26^o$ N, $121.49^o$ W).
     For each tower, the adjoint model is initialized at 00:00 UTC on March 31, 2018. We add an adjoint forcing of 1 ppm $CO_2$
at that time to the model grid cell closest to the tower location and the intake height (475m at WKT and 483m at WGC). The
     forcing is turned off for subsequent time steps and the adjoint model is run backward in time for 30 days, ending at 00:00 UTC
     on March 1, 2018. The resulting sensitivity of $CO_2$ at the WKT tower to the model's $CO_2$ mixing ratio, which is 3-dimensional,





is shown as a column average in the left panel of Fig. 2. The right panel of Fig. 2 shows the observation footprint (sensitivity to the surface flux scaling factor) at the corresponding times. The figures show that the sensitivity to the initial $CO_2$ is highest

and concentrated closest to the tower site at the time closest to the measurement: 5-days. With the increasing length of the backward-in-time integration, the sensitivity spreads over a larger area and its magnitude decreases. After 30 days, the sensitivity to the initial condition has propagated across most of the northern hemisphere. Figure 2 also indicates that the variation in the sensitivity magnitude decreases with time. To examine this, we calculated the standard deviation ($\sigma$) of sensitivity for each day of the 30-day period (Figure 3). The triangles in Fig. 3 show that the magnitude of the standard deviation of sensitivity to

the $CO_2$ mixing ratio decreases rapidly with time for both towers. On the other hand, the sensitivity to the surface flux scaling factor (footprint) exhibits a different pattern from the sensitivity to the initial $CO_2$. As shown in Fig. 2, the footprint spread spatially but the near field to the tower maintains a much higher magnitude than the far fields. By the end of 30 days, the footprint of WKT tower covers almost the entire northern hemisphere, with the area north and northwest of the tower within the conterminous United States exhibiting a much higher magnitude than the more distant area. The circles in Fig. 3 indicate

that the standard deviation of the footprint increase with time, but the rate of increase diminishes substantially after about 10 to 15 days. The finding suggests that extending the adjoint model integration further backward in time will still result in changes to the footprint, but with a much-reduced change rate.

For comparison, in Figure 4 we plot the MPAS-$CO_2$ adjoint model-calculated 10-day footprints in the CT-L geographic

domain. The figure reveals that the MPAS-$CO_2$ adjoint model-calculated WKT tower footprint spans most of the western and northwestern United States, with the highest sensitivity in Texas, Missouri, Iowa, Kansas, and Nebraska. Additionally, the footprint extends to a substantial area over the northeastern Pacific Ocean. The spatial pattern of the CT-L calculated footprint (Fig. 4c) is similar to that from the MPAS-$CO_2$ adjoint model, but it is visibly less continuous. Fig. 4 also shows that the MPAS-CO2 adjoint model-calculated footprint for the WGC tower covers northern California, southwestern Washington, and a portion of

the northeastern Pacific Ocean. The CT-L-calculated footprint exhibits a similar spatial pattern and magnitude. Overall, both the MPAS-$CO_2$ adjoint model and CT-L provide valuable information on the sensitivity of atmospheric $CO_2$ measurements to the surface flux: similar spatial patterns although with some differences due to resolution and the Lagrangian/Eulerian framework difference.

In the second set of experiments, we compare CT-L and MPAS-$CO_2$ adjoint model footprints for a swath of OCO-2 $X_{CO_2}$ measurements. The ground track of the OCO-2 orbit used in the experiments is indicated by the blue color line in Fig. 1. This orbit crosses North America from the Caribbean Sea to Canada's Northwest Territories in a northward direction between 18:31 UTC and 18:48 UTC on June 30, 2106. Since OCO-2 $X_{CO_2}$ represents the column average of atmospheric $CO_2$, CT-L calculates $X_{CO_2}$ footprints at 14 discrete height levels, ranging from $50$ to $14,000$ meters above the ground. For each height level,

footprints are computed by placing a number of particles at that specific height. To ensure consistency with the CT-L approach, the MPAS-$CO_2$ adjoint model is configured to apply the adjoint forcing at the corresponding vertical levels within the model. This configuration allows for a direct comparison between the footprints calculated by the MPAS-$CO_2$ adjoint model and the





CT-L footprints.

In the top panel of Fig. 5, we present the footprints of a point located south of Jamaica in the Caribbean Sea ($17.82^oN$, $77.88^oW$) at 500 m above the surface. Both the MPAS-$CO_2$ adjoint model and CT-L footprints largely extend eastward over the Atlantic Ocean, indicating transport from the surface due to the influence of the easterly trade winds. Additionally, the MPAS-$CO_2$ adjoint model-calculated footprint includes a branch that crosses the equator and extends southeastward to the southern hemisphere between $30^oW$ and $40^oW$ longitude. This feature is not shown in the CT-L footprint due to its limited-

area domain. In the lower panel of Fig. 5, we present the footprint of the same location but at 10,000 m above the surface. Both the MPAS-$CO_2$ adjoint and CT-L footprints show a primarily counterclockwise extension, covering the Gulf of Mexico and Texas. Moreover, there is a second segment extending westward from Texas toward the west coast. Upon closer examination, we observe that the CT-L-calculated footprint has a higher magnitude than the MPAS-$CO_2$ adjoint model over the Gulf of Mexico, but a lower magnitude over the mid-Atlantic regions of the United States, including from Kentucky to the Carolinas.

The distinct patterns in both systems' footprints at different vertical levels (500m and 10,000m) indicate significant differences in horizontal and vertical transport patterns.

    Figure 6 shows the corresponding footprints for an OCO-2 $X_{CO_2}$ location in eastern Kentucky ($36.8^oN$,$82.9^oW$) for particles released at 2,000 and 4,500m above the ground, respectively. The footprint of 2,000m extends predominantly northward,

covering the Great Lakes region and part of the Canadian Shield. In comparison, the footprint for 4,500m is mostly directed to the west. Another notable difference is that the highest magnitude portion of the 2,000m footprint is in close proximity to the point, while the 4,500m footprint is not in proximity to the point at all. These differences between the two height levels are evident in both the MPAS-$CO_2$ adjoint model and CT-L calculated footprints. In Fig. 7, we show the footprint of an OCO-2 sounding location on the southwest coast of Hudson Bay, Canada ($56.96^oN$, $91.89^oW$) at 500m and 4,500m altitude. Both the

MPAS-$CO_2$ adjoint and CT-L footprints for 500m are generally confined near to the sounding location, indicating that local surface fluxes have a significant influence on the atmospheric $CO_2$ at 500m above the surface. In comparison, the footprint of 4,500m is located more than 2,000km northwestward, mostly covering Alaska; the particles move that far horizontally in the time it takes them to advect and mix 4,500m in the vertical. These findings emphasize the significant impact of vertical mixing on the spatial distribution of footprint at different altitudes, highlighting the unique patterns of horizontal and vertical transport

in each case.

    In additional MPAS-$CO_2$ adjoint model runs, we quantitatively compare the footprints of the entire OCO-2 track at each of the 14 height levels with the CT-L footprints. This comparison is conducted by performing a single MPAS-$CO_2$ adjoint model run to calculate the footprint at the end of the 10-day backward-in-time integration for each height level. We then compare these

resulting footprints with their CT-L counterparts. Figure 8 shows the comparison at 4 height levels: 500m, 2,000m, 5,500m, and 10,000 m above the surface. The figure reveals that the footprints calculated by the two systems have similar spatial patterns within the limited-area domain of CT-L. However, it is important to note that a substantial portion of the footprints extends



beyond the CT-L domain. For instance, the footprints of 2,000m and 5,500m levels have significant coverage over Russian Siberia, while the footprint of the 10,000m level extends from the eastern Pacific Ocean to northeastern and western China,

both of which are outside the CT-L model domain.

In order to compare the footprints from the two systems quantitatively, we aggregated the footprints onto a $2^o \times 3^o$ (lat×lon) grid within the area covered by the CT-L model domain for each of the 14 height levels. Figure 9 shows the comparison for each of the 14. In the figure, the CT-L calculated footprints are on the X-axis, and MPAS-$CO_2$ adjoint model calculated footprints

are on the Y-axis. The figure demonstrates that the agreement between the two systems is better for footprints at lower heights than at higher heights. Specifically, at lower heights ranging from 500m to 2,500m, the MPAS-$CO_2$ adjoint model calculated footprints tend to have a somewhat higher magnitude than CT-L. However, at higher heights from 4,500m to 14,000m, the footprints calculated by CT-L tend to be of much higher magnitude compared to the MPAS-$CO_2$ adjoint model. These differences in magnitude between the two systems could be attributed to various factors, including differences in model configurations,

spatial resolution, and treatment of vertical mixing processes. Previous studies have shown that Lagrangian models, such as CT-L, can sometimes have different vertical mixing behavior compared to Eulerian models, especially at high altitudes (Karion et al., 2019).

In a final experiment, we use the MPAS-$CO_2$ adjoint model to examine the impact of different vertical distributions on

footprint calculation. Two adjoint model simulations are conducted for the OCO-2 orbit that crosses South America and North America between 17:36 UTC and 18:13 UTC on August 23, 2016 (the red color track in Fig. 1). Both simulations have the same adjoint forcing of 1 ppm $X_{CO_2}$ added at 18:00 UTC on August 23, 2016, and running backward in time for 30 days. The key difference between the two simulations lies in the vertical distributions of the adjoint forcing. For the first simulation, we adopt profile 1, which is obtained by combining the $X_{CO_2}$ averaging kernel and pressure weight function (O'Dell et al.,

2018). In contrast, profile 2 prioritizes $X_{CO_2}$ information in the lower part of the troposphere (Figure 10). This experiment aims to highlight how these differences in vertical distribution impact the footprint calculation, leading to variations in flux estimation using variational assimilation. The results of this experiment will provide valuable insights to the importance of selecting appropriate vertical distribution when using the adjoint model for $CO_2$ flux estimation.

The top two panels of Fig. 11 show the footprints resulting from MPAS-$CO_2$ adjoint model simulations using the two distinct vertical distribution profiles for the adjoint forcing (Fig. 10). Although the two footprints may initially appear very similar, substantial differences become evident as shown in the bottom panel in Fig. 11. Specifically, the footprint calculated using Profile 1 exhibits lower magnitudes compared to that obtained using Profile 2 in most extratropical regions in both the North Hemisphere and Southern Hemisphere. Conversely, in most of the tropics, the footprint calculated using Profile 1 shows

higher magnitudes than for Profile 2. Profile 2, which allocates more adjoint forcing to the lower atmosphere and less to the upper atmosphere, appears to be more sensitive to the stronger convective transport of surface $CO_2$ flux in the tropics than in the extratropics. These convective transport differences can account for the observed variations in the footprints between the two





profiles. These findings underscore the critical importance of selecting an appropriate vertical distribution for the model-data difference when using an adjoint model during variational assimilation.

## 380   6   Conclusions

The MPAS-CO$_2$ system consists of forward, TL, and adjoint models that are built upon the variable-resolution capability of the compressible non-hydrostatic MPAS-A model (Skamarock et al., 2012). It promises to be a useful tool for carbon flux inverse modeling at the global and regional scales. The forward model of MPAS-CO$_2$ is documented by Zheng et al. (2021). In this paper, we focus on the development of its tangent linear and adjoint models. Through rigorous testing, we have confirmed
the correctness and accuracy of the newly developed MPAS-CO$_2$ TL and adjoint models. A key challenge in developing the adjoint model was efficiently accessing meteorological variables during the model's backward-in-time integration. We have successfully implemented a strategy that combines recalculation and storage of meteorological variables. This approach significantly reduces the memory requirement, making the adjoint model feasible for long simulations, which are often necessary for CO$_2$ inverse modeling.

The results of the sensitivity analysis using the newly developed MPAS-CO$_2$ adjoint model provide valuable insights for designing CO$_2$ data assimilation systems. The increasing homogeneity of the sensitivity to the initial atmosphere CO$_2$ mixing ratio with longer integration length highlights the importance of selecting an appropriate assimilation window length. The comparison of the CO$_2$ observation footprints between the MPAS-CO$_2$ adjoint model and the NOAA CT-L system demonstrates
good agreement, validating the accuracy of the adjoint model's footprint calculations. The comparison of OCO-2 X$_{CO_2}$ footprints reveals differences in sensitivity between the two systems at different altitudes. MPAS-CO$_2$ adjoint model-calculated footprints tend to have higher magnitudes at low altitudes and lower magnitudes at high altitudes compared to CT-L. These differences likely arise from variations in vertical transport between the two systems. Lastly, the sensitivity analysis using two different vertical distribution profiles for adjoint forcing highlights the importance of correctly mapping model-data difference
in X$_{CO_2}$ to the transport model's vertical levels.

In addition to being a powerful tool for sensitivity analysis, the adjoint model plays a critical role in CO$_2$ variational data assimilation. Our future research efforts will focus on integrating the forward and adjoint models of MPAS-CO$_2$ into such a system. This integration has the potential to bridge a significant gap by establishing an online Eulerian transport model-based
global variational assimilation system for CO$_2$ that targets high resolution in critical regions while at the same time avoiding the pitfalls associated with the lateral boundaries needed in regional-domain inversions.

**Code and data availability**

The MPAS-CO$_2$ TL and adjoint models v7.3 developed in this paper can be downloaded from the CERN-based Zenodo archive at https://doi.org/10.5281/zenodo.8226620. This includes the model source code, instructions for compilation, and
example script for running models. Instructions for how to compile and run the models are provided in the package. CarbonTracker CO$_2$ flux and posterior mixing ratio data can be obtained from the NOAA website: https://www.esrl.noaa.gov/ gmd/ccgg/carbontracker/download.php. CT-L footprints can be obtained from the NOAA website: https://gml.noaa.gov/ccgg/ carbontracker-lagrange/

**Author contributions**

TZ designed and developed the MPAS-CO$_2$ TL and adjoint models. XT generated the 120-480km VR mesh used in model simulations. TZ, SF, JS, and XT designed and carried out the model accuracy verification experiments. TZ, SF, DB, and MB designed the adjoint sensitivity analysis experiments. All authors contributed to writing the paper.

**Competing interests**

The authors declare that they have no conflict of interest.

**Acknowledgment**

We thank the MPAS development team for making their code available to the public; We thank Dr. Bill Skamarock for his advice regarding the MPAS-A advection scheme. The model developments and simulations were carried out using Michigan State University High-Performance Computing Center (HPCC). PNNL is operated by the US Department of Energy by the Battelle Memorial Institute under contract DE-A05-76RL01830.

**7  Financial support**

This research has been supported by the National Aeronautics and Space Administration Carbon Monitoring System (CMS) grant no. 80HQTR21T0069.



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



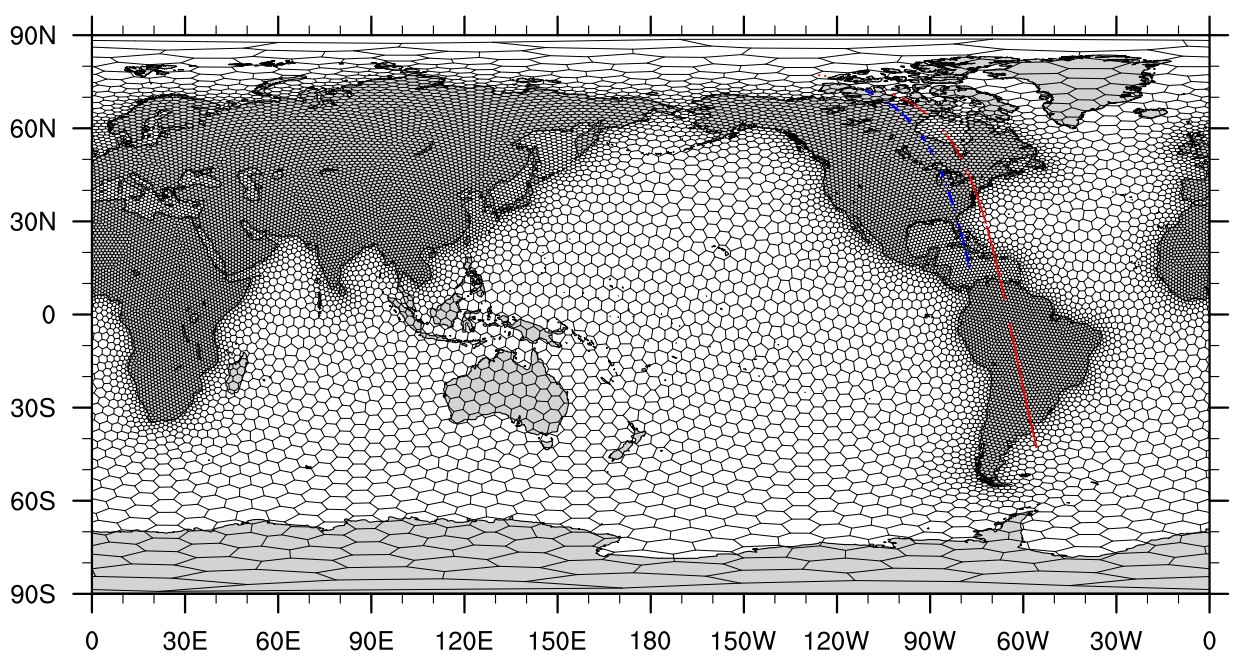

**Figure 1.** An MPAS-CO$_2$ global variable resolution mesh ranging from ∼120km over most of land regions to ∼480km over oceans.Also shown in the figures are the ground tracks of two OCO-2 orbits, which are used for the adjoint sensitivity studies described in Section 5. The blue-colored ground track crosses North America from the Caribbean Sea northward between 18:31 UTC and 18:48 UTC on June 30, 2016.The red-colored ground track crosses from South America to North America between 17:36 UTC and 18:13 UTC on August 23, 2016.



**Figure 2.** The column average of sensitivity of $CO_2$ at the WKT tower at 00:00 UTC on March 31 2018 to $CO_2$ mixing ratio (units: ppm/ppm) at 5, 10, 20, and 30 days backward in time (a)-(d). The WKT tower ($31.3149^{o}$N, $97.3269^{o}$W) measurements used here are taken 457m above the ground level and labeled by the red color cross in the figures of the left column. Figures (e)-(h) are the sensitivity of $CO_2$ at the tower at the same time to the surface $CO_2$ flux scaling factor (footprint, units:ppm/$\mu$mol m$^{-2}$ s$^{-1}$) computed 5, 10, 20, and 30 days backward-in-time, respectively.



**Figure 3.** The variation of the standard deviation ($\sigma$) of sensitivity to the initial $CO_2$ and the sensitivity to the flux scaling factor (footprint) over time. The standard deviations were calculated from MPAS-$CO_2$ adjoint model simulations starting on 2018 March 31 at 00:00 UTC, running 30 days backward in time, and ending on 2018 March 1 at 00:00 UTC. The top panel (a) is for the WKT tower (457 magl) while the bottom panel (b) is for the WGC tower. In each figure, the triangles represent the standard deviation of sensitivity to the $CO_2$ mixing ratio field (units: ppm/ppm), and the circles represent the standard deviation of footprint (units:ppm/$\mu$mol m$^{-2}$ s$^{-1}$).



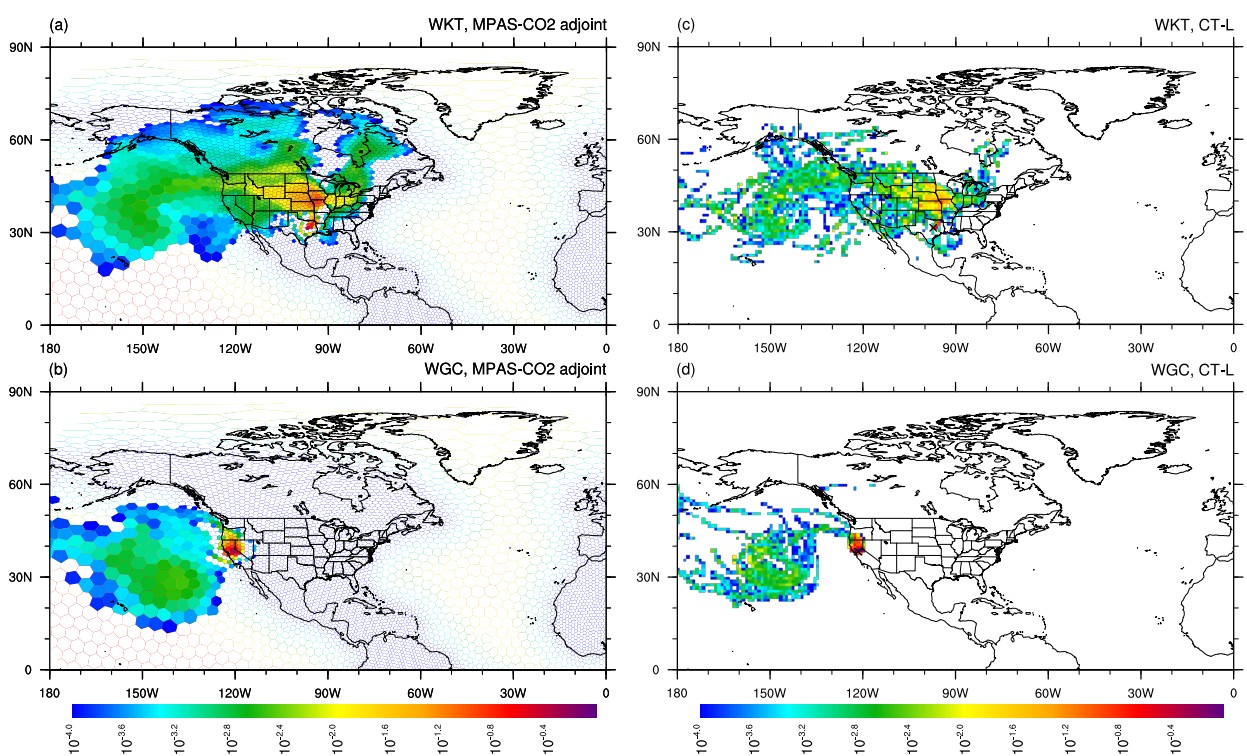

**Figure 4.** The 10-day backward in time $CO_2$ measurement footprint (units:ppm/$\mu$mol m$^{-2}$ s$^{-1}$) given by two tall towers: WKT and WGC. The figures on the top panel are the footprint of the WKT tower calculated using the MPAS-$CO_2$ adjoint model (a) and CT-L (c). The figures on the bottom panel are the footprint of the WGC tower calculated by the MPAS-$CO_2$ adjoint model (b) and CT-L (d). The location of the towers is marked by the black crosses in the figure on the right panel.



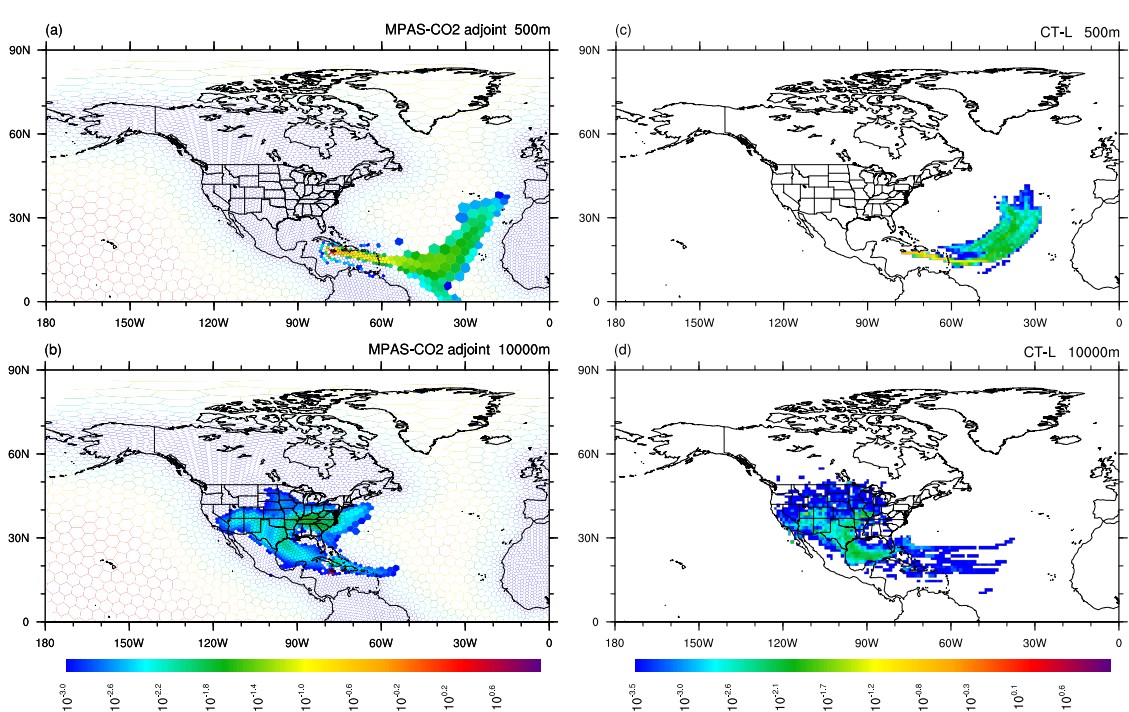

**Figure 5.** Comparison of footprints calculated by MPAS-CO$_2$ adjoint model and CT-L at a sounding location (red crosses, left panels) along the OCO-2 ground track in Fig 1(blue color). The footprints are calculated at two different heights: 500m (top panel) and 10,000m (bottom panel), and 10 days backward in time.





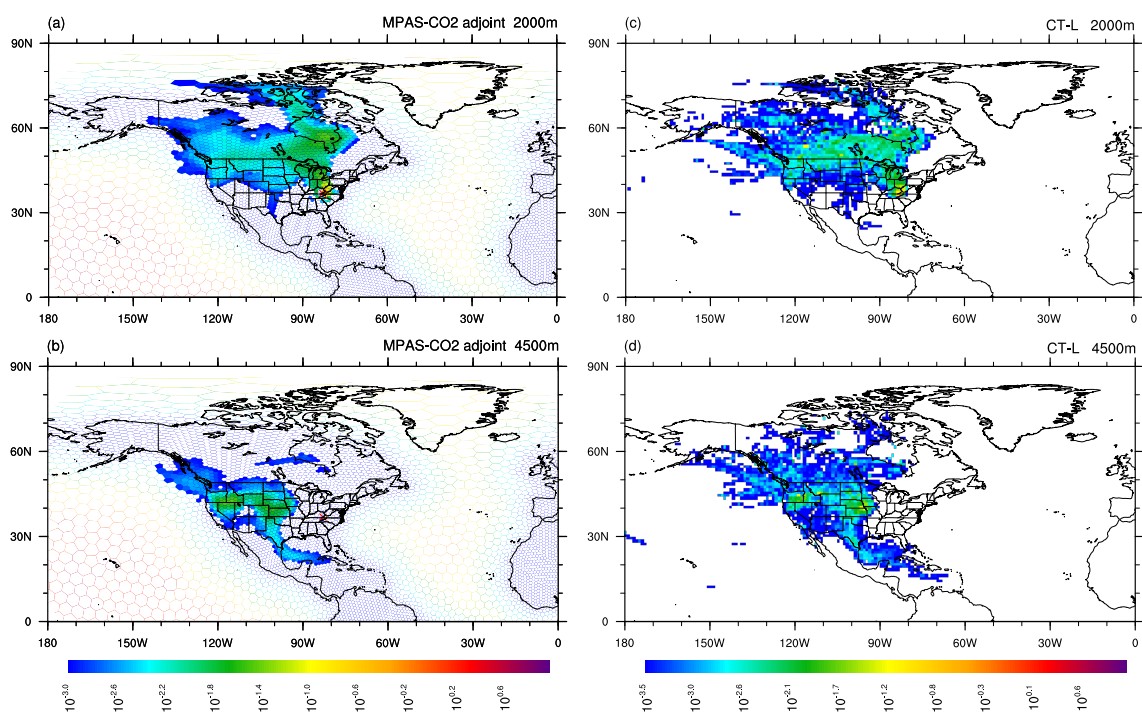

**Figure 6.** Same as Figure 5, except for a different OCO-2 sounding location and heights (2,000m in the top panel and 4,500m in the bottom panel).





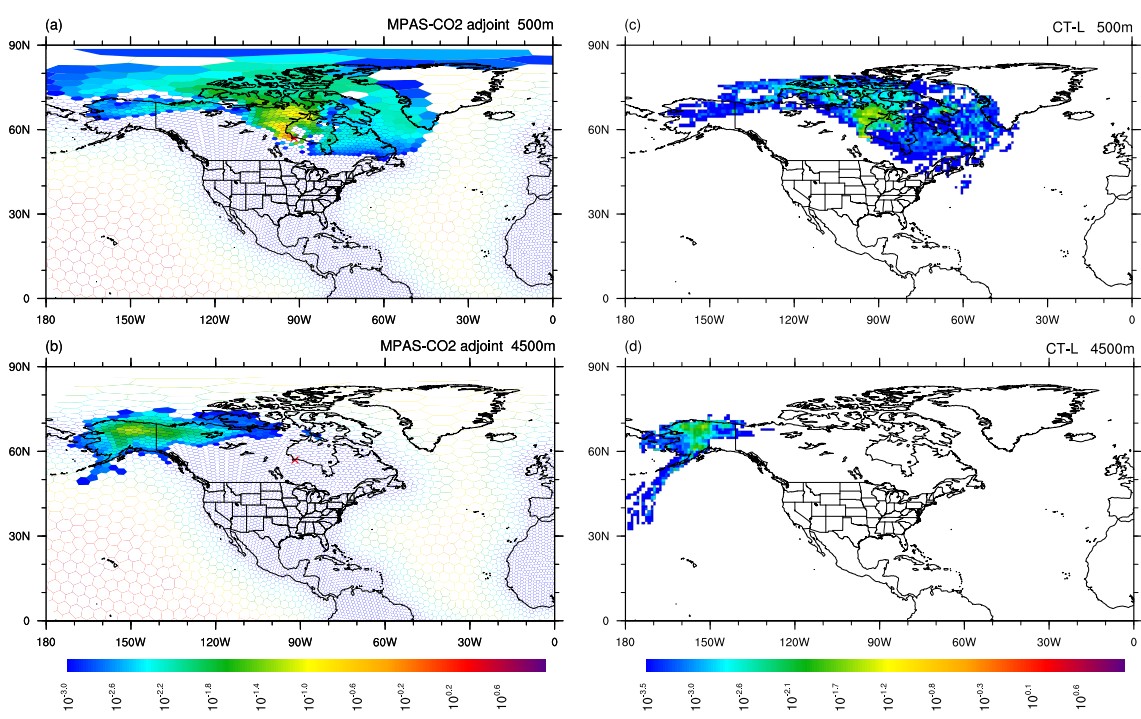

**Figure 7.** Same as Figure 5, except for a different OCO-2 sounding location and heights (500m in the top panel and 4,500m in the bottom panel).





**Figure 8.** The footprint of the OCO-2 ground track shown in Fig. 1 (with blue color) calculated by the MPAS-CO$_2$ adjoint model (left panel) and by CT-L (right panel). The footprints are calculated by placing the adjoint forcing (for the MPAS-CO2 adjoint model) or releasing particles (for CT-L) at four different height levels above the ground: 500m, 2000m, 5500m, and 10000m. The footprints are computed for 10 days backward in time.



**Figure 9.** Comparison of the OCO-2 groundtrack footprints from the MPAS-$CO_2$ adjoint model and CT-L after 10 days integration backward-in-time. For each of 14 height levels, the values of the footprints (units:ppm/$\mu$mol m$^{-2}$ s$^{-1}$ ) are extracted as the average value of $2°\times3°$ boxes within the range of the CT-L spatial domain (10-80°N, 180-10°W).



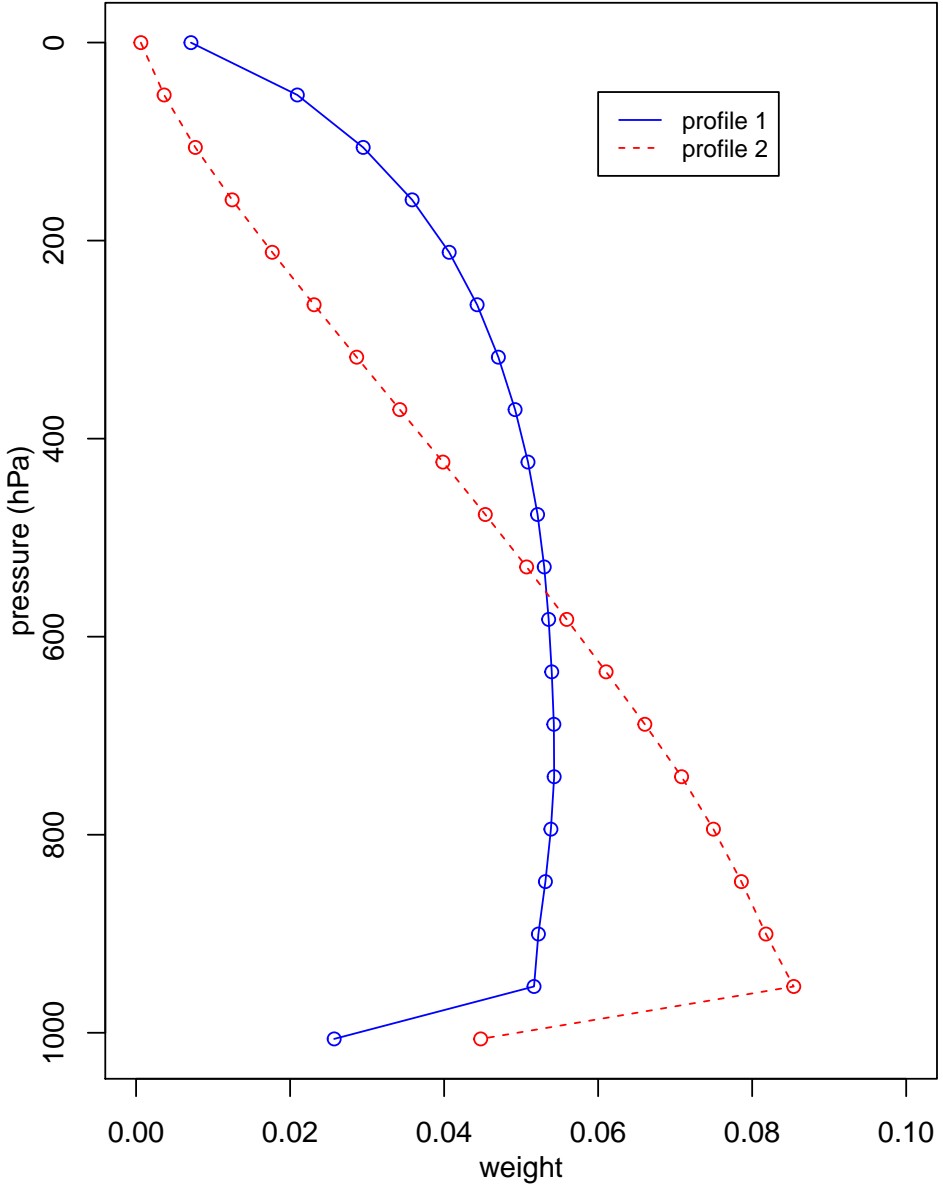

**Figure 10.** Two different profiles for vertically distributing a unit (ppm) of $X_{CO_2}$. Profile 1 is determined by OCO-2 $X_{CO_2}$ averaging kernel and pressure weight functions. Profile 2 is based on a redistribution of Profile 1 that gives more weight towards $CO_2$ in the lower troposphere than in the upper part of the atmospheric column. The circles of profiles are on the twenty pressure levels of OCO-2 $X_{CO_2}$ pressure weight function. Both profiles integrate to unity.

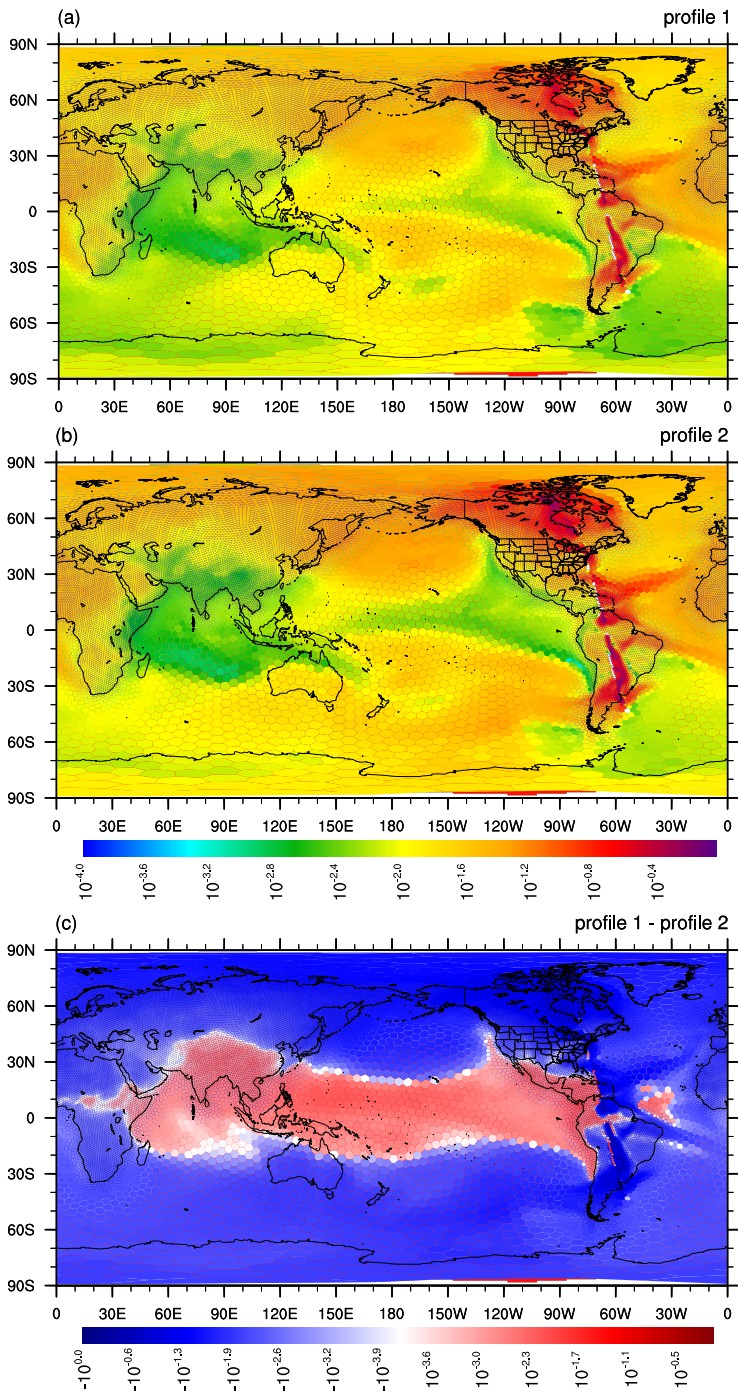

**Figure 11.** MPAS-CO$_2$ adjoint model-calculated footprints (units:ppm/$\mu$mol m$^{-2}$ s$^{-1}$) obtained after 30 days of backward-in-time integration starting on August 23, 2016 at 18:00 UTC (the time of the OCO-2 measurement). The top figure is obtained when using Profile 1 (Fig. 10) to vertically distribute 1 ppm of adjoint forcing. The middle figure is the footprint using Profile 2. The bottom figure is the difference in footprint between the two profiles.





**Table 1.** Results of the correctness check for the newly-developed MPAS-$CO_2$ tangent linear model. The results are from 1-month integration (from 2018-10-01 00:00 UTC to 2018-11-01 00:00 UTC) of the forward and tangent linear models using the 120-480 km global variable-resolution mesh (Fig. 1). The terms in the table refer to Eq. (9).

| $\alpha$ | $\| \mathcal{M}(\mathbf{x}_0,(1+\alpha)\mathbf{k}) - \mathcal{M}(\mathbf{x}_0,\mathbf{k}) \|$ | $\| \mathbf{M}(0,\alpha\mathbf{k}) \|$ | $\| \mathcal{M}(\mathbf{x}_0,(1+\alpha)\mathbf{k}) - \mathcal{M}(\mathbf{x}_0,\mathbf{k}) \| \, / \, \| \mathbf{M}(0,\alpha\mathbf{k}) \|$ |
|---|---|---|---|
| $1.0 \times 10^3$ | $2.07316571683768 \times 10^1$ | $2.07316571683768 \times 10^1$ | 1.0 |
| $1.0 \times 10^2$ | $2.07316571683768 \times 10^{-1}$ | $2.07316571683768 \times 10^{-1}$ | 1.0 |
| $1.0 \times 10^1$ | $2.07316571683768 \times 10^{-3}$ | $2.07316571683768 \times 10^{-3}$ | 1.0 |
| $1.0$ | $2.07316571683769 \times 10^{-5}$ | $2.07316571683768 \times 10^{-5}$ | 1.0 |
| $1.0 \times 10^{-1}$ | $2.07316571683765 \times 10^{-7}$ | $2.07316571683768 \times 10^{-7}$ | 0.99999999999998 |
| $1.0 \times 10^{-2}$ | $2.07316571683799 \times 10^{-9}$ | $2.07316571683768 \times 10^{-9}$ | 1.00000000000015 |
| $1.0 \times 10^{-3}$ | $2.07316571683735 \times 10^{-11}$ | $2.07316571683768 \times 10^{-11}$ | 0.99999999999984 |
| $1.0 \times 10^{-4}$ | $2.07316571692815 \times 10^{-13}$ | $2.07316571683768 \times 10^{-13}$ | 1.00000000004364 |





**Table 2.** Results of the correctness check for the newly-developed adjoint model of MPAS-CO$_2$. All simulations are of the 120-480km variable-resolution mesh (Fig. 1). The LHS and RHS in the table refer to Eq. (11).

| Integration length | LHS | RHS | (LHS-RHS)/LHS |
|---|---|---|---|
| $\Delta\mathbf{k} = 10^{-1}\mathbf{1}$ and $\Delta\mathbf{x} = \mathbf{M}(0, \Delta\mathbf{k})$ | | | |
| 7-day | $1.436630106778291 \times 10^{-7}$ | $1.436630106778298 \times 10^{-7}$ | $-5.158974640379662 \times 10^{-15}$ |
| 31-day | $2.073165716837682 \times 10^{-7}$ | $2.073165716837683 \times 10^{-7}$ | $-2.553561385538706 \times 10^{-16}$ |
| $\Delta\mathbf{k} = 10^{-1}\mathbf{1}$ and $\Delta\mathbf{x} = \mathcal{M}_{14d}(\mathbf{x_0}, \mathbf{1})$ | | | |
| 7-day | $2.273936055720336 \times 10^{-5}$ | $2.273936055720344 \times 10^{-5}$ | $-3.421966688503031 \times 10^{-15}$ |
| 31-day | $7.640482494092126 \times 10^{-5}$ | $7.640482494092106 \times 10^{-5}$ | $2.660668452525361 \times 10^{-15}$ |





**Table 3.** The computational costs for a 30-day simulation of MPAS-CO$_2$ at $\sim$120km quasi-uniform resolution and at a variable resolution ranging from $\sim$120 km to $\sim$480 km. Computational costs are shown for the forward, tangent linear, and adjoint models. All simulations are conducted using 128 AMD EPYC 7H12 2.595 GHz processors running in parallel.

| Model | Resolution (km) | Cost (min) | Time step (second) |
|---|---|---|---|
| Forward | 120 | 45 | 720 |
| | 120-480 | 20 | 720 |
| Tangent linear | 120 | 48 | 720 |
| | 120-480 | 22 | 720 |
| Adjoint | 120 | 48 | 720 |
| | 120-480 | 22 | 720 |