# Peer review of "Development of the tangent linear and adjoint models of the global online chemical transport model MPAS-CO2 v7.3"

_Geoscientific Model Development, 2023_

## Referee Comment (RC1)

**General comments**

The paper is well-structured and concise. The paper presents a useful tool to perform sensitivity analysis, and an essential building block to create a novel inverse modelling framework to be used in the future. The global variable-resolution mesh of the forward model is a promising tool to keep computational costs limited in this future inverse modelling framework, while avoiding issues related to lateral boundary conditions. The validation of the Tangent Linear (TL) and adjoint models is described in detail. The section on sensitivity could use some additional information on employed resolutions and employed initial meteorological conditions. The language is generally fluent and precise, and the authors refer extensively to existing literature. The introduction is also concise and to the point, and equations are well explained.

I have some specific comments, which will hopefully prove useful to further improve the manuscript.

**Specific comments**

- Line 11: '… satellite-derived column-averaged'. 'satellite-derived column-averaged $CO_2$ mixing ratio'?
- Line 15: Loss of information is mentioned here due to CT-L's simulation length and spatial domain limitations. In Section 5, I however only see simulations in which the simulation length is chosen identical to the one from the MPAS adjoint. So, aren't it only the spatial domain limitations that are relevant here?
- Line 49: This sentence reads somewhat strange, MPAS-CO2 addresses the limitation by being based on MPAS-A? The sentence coming after that is very clear though.
- Line 70: in equation 1, what is the meaning of the subscript cu in the last term? It is not immediately clear to me from the description. Maybe this can be shortly explained in the text if you think that is relevant?
- Equations 3 and 4: The statements $F_{bl} = \frac{\partial q_{co_2}}{\partial t}$ and $F_{cu} = \frac{\partial q_{co_2}}{\partial t}$ are slightly confusing to me. I assume that e.g. F_bl equals the local co2 tendency when only the PBL contribution is taken into account, not the full CO2 tendency. When applying the equations as they are written now consistently, it would mean that $\frac{\partial q_{co_2}}{\partial t}$ = F_bl = F_cu, which is presumably not what was intended. Maybe this can be made slightly more clear?
- Next to the above, From eq 1 it is clear that F_bl has the same units as $\frac{\partial(\tilde{\rho} q_{co_2})}{\partial t}$. It is not really clear to me how F_bl can have the units of $\frac{\partial q_{co_2}}{\partial t}$ (as follows from eq. 3) and $\frac{\partial(\tilde{\rho} q_{co_2})}{\partial t}$ at the same time, unless maybe $\tilde{\rho}$ is dimensionless? Adding units to the explanation of the variables could make things more clear.
- Equation 3: you are explaining what all the variables are that are used in equations 1, 2 and 3, but you do not explain the flux term $\overline{(w'q'_{co_2})_h}$ in equation 3.
- Line 102: 'e is 2-dimensional'. I assume 2-dimensional in space, but doesn't it have a time dimension as well?
- Line 117: What is meant with 'an emission driver of the forward model'?
- Equation 9: Maybe you can specify the meaning of the '||' symbols? I assume it is the 'length of vector' operator. The use of this operator in the equation however raises a question to me: Imagine a hypothetical case where the tangent linear model would contain errors. In that case, the vector

calculated using the TL model (M(0,αk), as in the denominator of equation 9) will be different from the vector calculated in the numerator of equation 9. However, this does not always imply that also the **length** of the vector M(0,αk) will be different from the length of the vector in the numerator. Thus, I would think that in this specific case with errors in the TL, equation 9 could still be satisfied, and thus does not guarantee correctness of the TL. Shouldn't this test be performed for each element of the TL output vector separately (thus for each mixing ratio in the grid at time t), thereby avoiding the need to work with the 'length of a vector' operator (see also eq. 6 of Errico 1997 and eq. 25 of Bosman and Krol 2023)? What is your opinion on this?

- Line 149 : I understand the reasoning that there would be no need to test sensitivity to x0, since the calculation of the sensitivity of x_t with respect to the flux scaling factor k involves the TL code of the CO2 emission driver in addition to the TL code of all the CO2 transport processes. But still, aren't the derivatives specifically to x0 untested in this way?

- Equation 10: About the use of $\Delta\hat{x}_t$ in the right-hand-side of the equation: Does the adjoint always 'ingest' the mixing ratios at forecast time, or can it also 'ingest' a time-array of observations between t0 and the forecast time (as is done in 4d-VAR)? In that case equation 10 is perhaps not a complete representation of the adjoint model

- Line 187-188: Reading these lines, it sounds to me as if the TL model does not require meteorological fields to run. Is that indeed true, and if so, why is this? If it does require meteorological fields (as I expect from lines 236-237), perhaps the sentence can be written slightly more clear.

- Line 190: '…during the adjoint model's forward sweep…'. Isn't it rather a forward model run during which the meteorological fields can be saved, instead of a 'forward sweep' of the adjoint model? Perhaps I am just used to a slightly different system of using an adjoint.

- Line 206: 'adjoint variable of flux scaling factor output from the adjoint model integration' Do you mean adjoint variable of flux scaling factor, which is output from the adjoint model integration from forecast time backward to the initial time? Perhaps this can be written slightly differently to make it more clear?

- Line 217: 'the agreement between the LHS and RHS of Eq. (11) is about −5.15×10−15'. Perhaps you can mention somewhere in this paragraph that it is not exactly zero due to (machine) rounding errors.

- Line 217: −5.16×10−15 instead of −5.15×10−15?

- Line 222: As mentioned on line 203, equation 11 should be satisfied for any combination of Δx and Δk. Therefore, I do not really understand why you refer to equation 9 here to justify your choice.

- Line 230: The forward model is not validated in this paper, I assume this is done in the Zheng et al. (2021) reference. So perhaps the sentence could be slightly reformulated.

- Line 236: 'the TL and adjoint model simulations using the same configuration take approximately 10% longer, indicating that the majority of the computation time is used for integrating the meteorological fields'. This confuses me somewhat: Aren't the meteorological fields calculated during a forward model run as well?

- Line 262-263 'When a uniform time-invariant surface flux is used, the sensitivity to the surface flux scaling factor calculated by the MPAS-CO2 adjoint model is the observation footprint.' Isn't this only true when the flux is set to 1, otherwise it should still be multiplied with a fixed factor? And shouldn't S(k) be taken into account as well?

- Line 265: What function is chosen for S(k) in equation 6?

- Line 267: Where do the initial meteorological conditions (for generating the meteorological fields for the adjoint) come from?

- Line 270: I read here an intake height of 475 m for the WKT tower, but the captions of Figure 2 and Figure 3 mention 457 m…

- Caption of figure 2: The units of the sensitivity of CO2 at the tower to the surface CO2 flux scaling factors are given here as ppm/μmol m−2 s−1. However, given equation 6, I assume the scaling factors to be dimensionless. Because of that, one expects the units of the sensitivity to be ppm instead.

However, because we use a fixed flux of 1.0 µmol m−2 s−1, the sensitivity to the flux scaling factor is equal to the sensitivity to the scaled CO2 flux (e in Eq. 6). Perhaps it might be good to clarify the origin of these somewhat counterintuitive units for the sensitivity to the surface CO2 flux scaling factor.

- Figure 2: In all of the panels, there is a peculiar 'red blob' present in Antarctica. As it doesn't look like this has a physical origin, it might need some explanation/correction. Also in e.g. Figure 8 I notice this.
- Caption Figure 3: the height of only one of the towers is given here. Perhaps better to give either none or both.
- Line 277: '…indicates that the variation in the sensitivity magnitude decreases with time' Do you mean spatial variation or temporal variation? Maybe good to state this explicitly.
- Line 279: 'The triangles in Fig. 3 show that the magnitude of the standard deviation of sensitivity to the CO2 mixing ratio decreases rapidly with time for both towers'. I find this statement somewhat confusing, given that time runs backward in the figure, and thus σ actually increases over (forward) time. Perhaps you could find an elegant way to make this clear. The same holds for line 285.
- Line 286: Interesting finding, but how can we know for sure that a small change rate of the σ of the footprint also means a small change of the footprint itself?
- Line 294: Oregon and Nevada are not mentioned?
- Line 306: How well do the levels of the two models correspond? Is it exact?
- Line 318: The comparison would be easier when the left and right panels in figure 5 would use the same color scale, now there is a notable difference between the scales. This is also the case in Figure 6 and 7.
- Line 330: 'Both the MPAS-CO2 adjoint and CT-L footprints for 500m are generally confined near to the sounding location'. Looking at the figure I would say there is still quite some sensitivity to further away regions as well.
- Line 338: Is it a single adjoint model run, or a single run for each height level? This is not immediately clear to me the way the sentence is written.
- Line 339: Am I interpreting it correctly, when I say that the sensitivity at a single height level in Fig. 8 is the sensitivity of the average mixing ratio along the OCO-2 track at the specified model height, whereby the average is taken horizontally over all cells that are part of the OCO-2 track? Perhaps it might be good to add a little more explanation at this point in the paper.
- At which resolutions are the CT-L and adjoint models run in Section 5? I suggest to include this information.
- Line 352: 'at higher heights from 4,500m to 14,000m, the footprints calculated by CT-L tend to be of much higher magnitude compared to the MPAS-CO2 adjoint model'. Looking at the figures, this only holds for a fraction of the footprints, definitely not all of them. Perhaps good to make this clear.
- Figure 9: Just a suggestion, but wouldn't it be interesting to include one or two simple metrics in the subfigures to numerically quantify the difference between the adjoint and CT-L? Now it is purely visual. Perhaps just the difference in the means and a measure for correlation, or maybe a (scaled) root mean squared difference or something… The discussion on page 12 could then also be based on these numbers.
- Caption Figure 9: '…extracted as the average value of 2°x3° boxes within the range of the CT-L spatial domain (10-80°N, 180-10°W)'.   However, at line 257  the longitude of the CT-L domain is given as '0°-180° W', which is different.
- Line 362: Is the adjoint forcing added to all the cells where the OCO-2 track passes? It is not mentioned where the forcing is added.
- Figure 10: To what extent do these 20 pressure levels coincide with MPAS-CO2 pressure levels?
- Line 375: 'Profile 2, …, appears to be more sensitive to the stronger convective transport of surface CO2 flux in the tropics than in the extratropics'.   Do you mean 'appears to be more sensitive than profile 1 to the stronger convective transport…'? Perhaps good to make this clear. Next to that, I don't understand where this statement comes from. In the figure I see that the footprints of Profile 2 are generally smaller than those of footprint 1 in the tropics, so shouldn't the statement be the opposite

of what you state now? If not, Perhaps you can explain your reasoning how you go from the information in the figure to the statement you make (also in the paper)?

- Line 377: 'These convective transport differences…' Convective transport differences between the 2 adjoint simulations? Given that the two adjoint simulations use the same meteorology etc., this doesn't read very clear to me, or maybe I don't understand it properly. Do you mean differences between the 2 footprints, originating from convective transport to higher levels of the atmosphere?
- Comparing line 398 with 354-355, it seems the given list of reasons for the differences has now been replaced by 'likely arise from variations in vertical transport'? Perhaps not fully consistent? Or is there a specific reason why you can narrow it down here? Please check also whether the related part in the abstract it is consistent with lines 354-355.
- A small note on the code and data availability section, for your information: GMD encourages the submission of code to perform calculations described in the text, see https://www.geoscientific-model-development.net/peer_review/review_criteria.html.
- Line 408: Perhaps mention that the forward model is included as well.

**Technical comments**

- Line 64: 'conclusion**s**'?
- Line 91: small typo, the word 'two' occurs twice
- Line 106: Why is x_t written as X_t here? Or is X_t another variable?
- Line 172: 'Eq. (9)' instead of (9)
- Line 214: I think the word 'ratio' is missing after 'mixing'
- Line 222: 'configuration' instead of 'configurations'?
- Caption Figure 1: Small typo in first sentence, 'A' instead of 'An'
- In the rightmost column of Table 1, it is perhaps better to use the same number of significant digits or decimal places for all lines, now there is a sudden 'jump' from 1.0 to $0.99999999999998$, while the precision of both numbers is actually the same or very similar.
- Caption figure 3: just a suggestion: maybe change the rather complex first sentence 'The variation of the standard deviation (σ) of sensitivity to the initial CO2 and the sensitivity to the flux scaling factor (footprint) over time' into something like 'The variation over time of the standard deviation (σ) of two quantities: the sensitivity to the initial CO2 and the sensitivity to the flux scaling factor (footprint)'.
- Line 281: small typo, 'the footprint spread…'
- Caption Figure 4: 'black crosses in the figures on the right panel' instead of 'black crosses in the figure on the right panel'?
- Line 303: Small typo, I assume '2106' should be 2016.
- Line 362 '…and running backward in time…' The grammar seems not fully correct in this sentence.

**References**

Bosman, P. J. M. and Krol, M. C.: ICLASS 1.1, a variational Inverse modelling framework for the Chemistry Land-surface Atmosphere Soil Slab model: description, validation, and application, Geosci. Model Dev., 16, 47–74, https://doi.org/10.5194/gmd-16-47-2023, 2023.

Errico, R. M.: What is an adjoint model?, Bulletin of the American Meteorological Society, 78(11), 2577‒2591, 1997.

---

## Author Response (AR1)

**Response to editor comments**

We thank the editor for coordinating the review and providing us with the comments In the following section, the editor's comments are in blue and our responses are in black.

I am very pleased to inform you about the outcome of my quick access review: Your manuscript fulfills all basic GMD requirements and is acceptable for publication in GMD discussions.

During my evaluation I found a few minor typos which may be corrected at a later stage of the review process:

- L91: two two different numerical schemes
Fixed.

- L157: M(0,$\alpha$k)); two closing brackets
Fixed.

- L225: agree to to about
Fixed.

- L374: North Hemisphere $\rightarrow$ Northern Hemisphere
Fixed.

**Response to reviewer 1**

We thank the reviewer for providing us with comprehensive and insightful comments. Through thoroughly addressing these comments, we have improved the quality of this manuscript. In the following sections, the reviewer's original comments are in blue and our responses are in black.

**General comments**

The paper is well-structured and concise. The paper presents a useful tool to perform sensitivity analysis, and an essential building block to create a novel inverse modeling framework to be used in the future. The global variable-resolution mesh of the forward model is a promising tool to keep computational costs limited in this future inverse modelling framework, while avoiding issues related to lateral boundary conditions. The validation of the Tangent Linear (TL) and adjoint models is described in detail. The section on sensitivity could use some additional information on employed resolutions and employed initial meteorological conditions. The language is generally fluent and precise, and the authors refer extensively to existing literature. The introduction is also concise and to the point, and equations are well explained.

We appreciate the reviewer's comprehensive and constructive comments.

Additional information regarding simulation resolution and initial meteorological conditions for the adjoint sensitivity experiments (Section 5) have been added. Please see our responses to specific comments 23 (meteorological initial conditions) and 37 (adjoint model simulation horizontal resolution).

I have some specific comments, which will hopefully prove useful to further improve the manuscript.

**Specific comments**

1)- Line 11: '... satellite-derived column-averaged'. 'satellite-derived column-averaged CO2 mixing ratio'?
Thanks. The phrase has been corrected to "satellite-derived column-averaged $CO_2$ mixing ratio" (*line* 11).

2)- Line 15: Loss of information is mentioned here due to CT-L's simulation length and spatial domain limitations. In Section 5, I however only see simulations in which the simulation length is chosen identical to the one from the MPAS adjoint. So, aren't it only the spatial domain limitations that are relevant here?
Agreed. The sentence has been revised to "Moreover, this comparison highlights the substantial loss of information in the atmospheric $CO_2$ observations due to CT-L's spatial domain limitation." (*line 15*)

3)- Line 49: This sentence reads somewhat strange, MPAS-CO2 addresses the limitation by being based on MPAS-A? The sentence coming after that is very clear though.
Thanks for pointing this out. The sentence has been revised to "As a global online chemical transport model, MPAS-$CO_2$ avoids the lateral boundary condition problem." (*line 49*)

4)- Line 70: in equation 1, what is the meaning of the subscript cu in the last term? It is not immediately clear to me from the description. Maybe this can be shortly explained in the text if you think that is relevant?
Thanks for pointing this out. The sentence has been revised to " The second ($F_{bl}$) and third ($F_{cu}$) terms of RHS represent the contribution from the vertical mixing by the planetary boundary layer (PBL) and cumulus convective transport parameterizations, respectively." (*line 75*). This makes it clear that "cu" subscript represents the convective transport of $CO_2$ by the cumulus parameterization.

5) - Equations 3 and 4: The statements $F_{bl} = \frac{\partial q_{bl}}{\partial t}$ and are slightly confusing to me. I assume that e.g. $F_b l$ equals the local co2 tendency when only the PBL contribution is taken into account, not the full $CO_2$ tendency. When applying the equations as they are written now consistently, it would mean that $\frac{\partial q_{co2}}{\partial t} = F_{bl} = F_{cu}$, which is presumably not what was intended. Maybe this can be made slightly more clear?
Thanks for pointing this out. A mistake was made when we tried to summarize the detailed forward model description (Zheng et al. (2021)) into a brief version in this manuscript. This has been corrected as follows:
Eq.(3) has been corrected from

$$F_{bl} = \frac{\partial q_{co_2}}{\partial t} = \frac{\partial}{\partial z}\left[K_h(\frac{\partial q_{co_2}}{\partial z}) - \overline{(w'q'_{co_2})_h}\,(\frac{z}{h})^3\right]$$

to:

$$\left[\frac{\partial q_{co_2}}{\partial t}\right]_{bl} = \frac{\partial}{\partial z}\left[K_h(\frac{\partial q_{co_2}}{\partial z}) - \overline{(w'q'_{co_2})_h}\,(\frac{z}{h})^3\right]$$

The text also has been revised to make it clear that

- The term $\left[\frac{\partial q_{co_2}}{\partial t}\right]_{bl}$ of Eq. (3) is the $CO_2$ tendency contribution from the PBL parameterization before being coupled with the dry-air density.

- The term $\left[\frac{\partial q_{co_2}}{\partial t}\right]_{bl}$ of Eq. (3) is coupled with dry-air density $\tilde{\rho}$ to form the term $F_{bl}$ of Eq. (1). (*line 85*)

Similarly, Eq. (4) has been corrected to:

$$\left[\frac{\partial q_{co_2}}{\partial t}\right]_{cu} = \frac{(M_u + M_d)}{\rho A}\frac{\partial q_{co_2}}{\partial z} + \frac{M_{ud}}{M}(q_{co_2}^u - q_{co_2}) + \frac{M_{dd}}{M}(q_{co_2}^d - q_{co_2})$$

, and $\left[\frac{\partial q_{co_2}}{\partial t}\right]_{cu}$ is coupled with dry-air density $\tilde{\rho}$ to form $F_{cu}$ of Eq. (1) (*line 94*)

Please note that the above corrections also respond to the reviewer's comment 6 (below).

6) - Next to the above, From eq 1 it is clear that $F_{bl}$ has the same units as $\frac{\partial \rho \tilde{q_{co2}}}{\partial t}$ . It is not really clear to me how $F_{bl}$ can have the units of $\frac{\partial q_{co2}}{\partial t}$ (as follows from eq. 3) and $\frac{\partial \rho \tilde{q_{co2}}}{\partial t}$ at the same time, unless maybe is dimensionless? Adding units to the explanation of the variables could make things more clear.
Thanks for pointing this out. Please see our response to the reviewer's comment 5.

7) - Equation 3: you are explaining what all the variables are that are used in equations 1, 2 and 3, but you do not explain the flux term $\overline{(w'q'_{co_2})}_h$ in equation 3.
Thanks for pointing this out. The following sentence has been added in the revision to explain the term $\overline{(w'q'_{co_2})}_h$ : "The second term of the RHS in the square bracket of Eq. (3) represents the contribution from $CO_2$ entrainment flux at the inversion layer." (*line 84*)

8) - Line 102: 'e is 2-dimensional'. I assume 2-dimensional in space, but doesn't it have a time dimension as well?
Thanks for pointing it out. The sentence has been revised to "While both $\mathbf{x}_0$ and $\mathbf{x}_t$ are 3-dimensional in space, $\mathbf{e}$ is 2-dimensional in space, indicating that $CO_2$ flux is applied only to the model's surface cells." (*line 107*)

9) - Line 117: What is meant with 'an emission driver of the forward model'?
The sentence has been revised to "We implemented Eq. (6) in the forward model in a way that allows the flexibility of choosing the temporal frequency of the flux scaling factors." (*line 123*)

10) Equation 9: Maybe you can specify the meaning of the '| |' symbols? I assume it is the 'length of vector' operator. The use of this operator in the equation however raises a question to me: Imagine a hypothetical case where the tangent linear model would contain errors. In that case, the vector calculated using the TL model $M(0, \alpha k)$, as in the denominator of equation 9) will be different from the vector calculated in the numerator of equation 9. However, this does not always imply that also the length of the vector $M(0, \alpha k)$ will be different from the length of the vector in the numerator. Thus, I would think that in this specific case with errors in the TL, equation 9 could still be satisfied, and thus does not guarantee correctness of the TL. Shouldn't this test be performed for each element of the TL output vector separately (thus for each mixing ratio in the

grid at time t), thereby avoiding the need to work with the 'length of a vector' operator (see also eq. 6 of Errico 1997 and eq. 25 of Bosman and Krol 2023)? What is your opinion on this?

This is a really interesting question. First, yes. the symbol "$|| \ ||$" in Eq. (9) of our manuscript is indeed the "length of vector operator" ($L^2$ norm operator).

$$\frac{dH(\mathbf{x}_m, p)_i}{d\mathbf{x}_m} \cdot \Delta\mathbf{x}_m \approx \frac{H(\mathbf{x}_m + \alpha\Delta\mathbf{x_m}, p)_i - H(\mathbf{x}_m, p)_i}{\alpha}$$

Above is Eq. (25) of Bosman and Krol 2023 (BK23 afterward), where $H(\mathbf{x}_m + \alpha\Delta\mathbf{x_m}, p)_i$ is the i$^{th}$ output of the forward model, and $\frac{dH(\mathbf{x}_m, p)_i}{d\mathbf{x}_m}$ is the i$^{th}$ output of the TL model. The forward model of BK23, Chemistry Land-surface Atmosphere Soil Slab (CLASS) is a column model (1-dimensional in space), which has a relatively small number of outputs, including those listed in Table 3 of BK23. The small number of model outputs of the CLASS model makes it possible to use Eq. (25) to verify the TL model's correctness. In comparison, MPAS-CO$_2$, which is a 3-dimensional model, has a much larger number of outputs. For instance, the 120-480 km variable resolution mesh (see Table 1 of the manuscript) has 15,988 horizontal cells and 55 vertical levels, meaning the number of its CO$_2$ mixing ratio output is $15,898 \times 55 = 874,390$. Such a large number of model outputs makes it impractical to apply Eq. (25) of BK23 to the MPAS-CO$_2$ TL model verification.

Eq. (9) in our manuscript is a standard test for TL model correctness, which has been widely used for 3-dimensional atmospheric dynamic models, such as in Tian and Zou (2020) and Zou et al. (1997). In addition, we examined the 3-dimensional field of difference between $\mathcal{M}(\mathbf{x_0}, (1 + \alpha)\mathbf{k}) - \mathcal{M}(\mathbf{x_0}, \mathbf{k})$ and $\mathbf{M}(0, \alpha\mathbf{k})$ for different $\alpha$ values. The results show that the values at all cells of the 3-dimensional fields are virtually zero (not shown). This rules out the possibility that the reviewer raised: the TL model contains errors but Eq. (9) is still satisfied.

11)- Line 149: I understand the reasoning that there would be no need to test sensitivity to $x_0$, since the calculation of the sensitivity of $x_t$ with respect to the flux scaling factor $k$ involves the TL code of the CO2 emission driver in addition to the TL code of all the CO2 transport processes. But still, aren't the derivatives specifically to x0 untested in this way?

In the MPAS-CO$_2$ model system, CO$_2$ fluxes are added to the surface cell's CO$_2$ mixing ratio at the model's every time step, including the initial time. This modification of CO$_2$ mixing ratio at the initial time is equivalent to a perturbation to $\mathbf{x}_0$, meaning that the sensitivity of $\mathbf{x}_t$ to the flux (through the flux scaling factor) includes the sensitivity to $\mathbf{x}_0$. Thus, the correctness of the TL and adjoint models with respect to the initial CO$_2$ mixing ratio ($\mathbf{x}_0$) is a necessary condition for the correctness of the models with respect to the flux scaling factor. In other words, if the TL and adjoint models contain any error regarding the calculation of the sensitivity $\frac{\partial\mathbf{x}_t}{\partial\mathbf{x}_0}$, the tests of Eqs. (9) and (10) would not succeed. Therefore, the tests conducted in the paper indeed prove that the TL and adjoint models are correctly coded in calculating $\frac{\partial\mathbf{x}_t}{\partial\mathbf{x}_0}$.

12)- Equation 10: About the use of in the right-hand-side of the equation: Does the adjoint always 'ingest' the mixing ratios at forecast time, or can it also 'ingest' a time-array of observations between t0 and the forecast time (as is done in 4d-VAR)? In that case equation 10 is perhaps not a complete representation of the adjoint model.

The adjoint model of MPAS-CO$_2$ is implemented to be able to ingest a time series of observations between the initial time ($t_0$) and the end of the simulation period. The observation ingestion time interval can be specified at the run time. To clarify this point, the sentence that describes Equation 10 has been revised to "Eq. (10) shows that starting with $\Delta\hat{\mathbf{x}}_t$ at the forecast time, the

MPAS-$CO_2$ adjoint model runs backward in time to the initial time while ingesting $CO_2$ observations along the way, resulting in the adjoint variable of $CO_2$ mixing ratio at the initial time ($\Delta\hat{\mathbf{x}}_0$), and the adjoint variable of the flux scaling factor ($\Delta\hat{\mathbf{k}}$)". (*line 191*)

13) - Line 187-188: Reading these lines, it sounds to me as if the TL model does not require meteorological fields to run. Is that indeed true, and if so, why is this? If it does require meteorological fields (as I expect from lines 236-237), perhaps the sentence can be written slightly more clearly. We agree with the reviewer that this segment needs to be written more clearly. This has been revised to as "The calculation of the $CO_2$ transport needs access to the meteorological fields at each time step. Since the forward and TL models both run forward in time, this access is straightforward. However, because the adjoint model runs backward in time, accessing the meteorological field is more challenging." (*line 196-198*)

14) - Line 190: '...during the adjoint model's forward sweep...'. Isn't it rather a forward model run during which the meteorological fields can be saved, instead of a 'forward sweep' of the adjoint model? Perhaps I am just used to a slightly different system of using an adjoint.
This possible confusion is due to the differences between the online and offline transport models and their adjoints. The adjoint of an offline transport model can use the meteorological fields saved in files by a previous forward model run. The adjoint of an online transport model, such as MPAS-$CO_2$, runs a forward sweep, which is the same process as the forward model and saves the meteorological fields in memory for the subsequent backward sweep to access. This is how the adjoint codes of online chemical transport models were implemented in Guerrette and Henze, 2015, Zheng et al 2018, and this manuscript.

15) - Line 206: 'adjoint variable of flux scaling factor output from the adjoint model integration' Do you mean adjoint variable of flux scaling factor, which is output from the adjoint model integration from forecast time backward to the initial time? Perhaps this can be written slightly differently to make it more clear?
Agreed. The sentence has been revised to "The first item of the RHS, $\mathbf{M}^T(\Delta\mathbf{x})$, represents the adjoint variable of flux scaling factor, which is an output from the adjoint model integration from the forecast time backward to the initial time." (*line 221*)

16) - Line 217: the agreement between the LHS and RHS of Eq. (11) is about $-5.15 \times 10^{-15}$ Perhaps you can mention somewhere in this paragraph that it is not exactly zero due to (machine) rounding errors.
Agreed. The following sentence has been added to the revision: "We note that this value is not exactly zero due to the machine rounding errors". (*line 232*)

17) - Line 217: $-5.16 \times 10^{-15}$ instead of $-5.15 \times 10^{-15}$ ?
Thanks for pointing it out. It has been corrected to $-5.16 \times 10^{-15}$. (*line 232*)

18) - Line 222: As mentioned on line 203, equation 11 should be satisfied for any combination of $\Delta x$ and $\Delta k$. Therefore, I do not really understand why you refer to equation 9 here to justify your choice.
Thanks for pointing this out. Yes, it should refer to Eq. (11). This has been corrected. (*line 238*)

19) - Line 230: The forward model is not validated in this paper, I assume this is done in the

Agreed. This segment of text has been revised as "This validation ensures the reliability and integrity of the entire MPAS-$CO_2$ model system since the forward model has already been validated in Zheng et al. (2021)." (*line 246*)

20) - Line 236: 'the TL and adjoint model simulations using the same configuration take approximately 10% longer, indicating that the majority of the computation time is used for integrating the meteorological fields'. This confuses me somewhat: Aren't the meteorological fields calculated during a forward model run as well?

Thanks for pointing this out. The sentence has been revised to "Both the TL and adjoint model simulations using the same configuration take approximately 10% longer than the forward model. This extra computation time for the TL and adjoint models is incurred by the execution of the TL/adjoint code of the $CO_2$ transport processes." (*line 252-253*)

21) - Line 262-263 'When a uniform time-invariant surface flux is used, the sensitivity to the surface flux scaling factor calculated by the MPAS-CO2 adjoint model is the observation footprint.' Isn't this only true when the flux is set to 1, otherwise it should still be multiplied with a fixed factor? And shouldn't S(k) be taken into account as well?

Thanks for pointing this out. The sentence has been revised to "When a spatially uniform time-invariant surface flux of underline{unity value} is used and $S(\mathbf{k}) = \mathbf{k}$ in Eq. (6), the sensitivity to the surface flux scaling factor calculated by the MPAS-$CO_2$ adjoint model is the observation footprint." (*line 280-281*)

22) - Line 265: What function is chosen for S(k) in equation 6?

For the MPAS-$CO_2$ adjoint model simulations conducted for comparison with CT-L footprints, $S(\mathbf{k}) = \mathbf{k}$ in Eq. (6). Please our response to the reviewer comment 21.

23) - Line 267: Where do the initial meteorological conditions (for generating the meteorological fields for the adjoint) come from?

The following sentences have been added to Section 5 to clarify the MPAS-$CO_2$ adjoint model simulation configuration including the initial meteorological conditions.

"Meteorological initial conditions for MPAS-$CO_2$ model simulations conducted in this section are from the ERA5 reanalysis (Hoffmann et al. 2019). Footprints calculated by the MPAS-$CO_2$ adjoint model are of the 120-480km variable resolution grid while the CT-L footprints are of $1° \times 1°$." (*line 286-288*)

24) - Line 270: I read here an intake height of 475 m for the WKT tower, but the captions of Figure 2 and Figure 3 mention 457 m.

Thanks for pointing this out. The intake height of WKT tower is 457 m. It has been corrected in the revision (*line 293*).

25) - Caption of figure 2: The units of the sensitivity of CO2 at the tower to the surface CO2 flux scaling factors are given here as ppm/$\mu$mol m$^{-2}$ s$^{-1}$. However, given equation 6, I assume the scaling factors to be dimensionless. Because of that, one expects the units of the sensitivity to be ppm instead, However, because we use a fixed flux of 1.0 $\mu$ mol$^{-2}$ s$^{-1}$, the sensitivity to the flux scaling factor is equal to the sensitivity to the scaled CO2 flux (e in Eq. 6). Perhaps it might be good to clarify the origin of these somewhat counterintuitive units for the sensitivity to the

surface CO2 flux scaling factor.

Agreed. The following sentence has been added to explain the footprint units.

"Because $\mathbf{S}(\mathbf{k}) = \mathbf{k}$ is used, Eq. (6) takes the form of $\mathbf{e} = \mathbf{k}\,\tilde{\mathbf{e}}$. The units of the $CO_2$ flux ($\mu$mol m$^{-2}$ s$^{-1}$) and the multiplicative nature of the flux scaling factor $\mathbf{k}$ determine the units of the adjoint variable $\Delta\hat{\mathbf{k}}$ (which represents observation footprints) to be ppm/($\mu$mol m$^{-2}$ s$^{-1}$)."(*line 283-285*)

26) - Figure 2: In all of the panels, there is a peculiar 'red blob' present in Antarctica. As it doesn't look like this has a physical origin, it might need some explanation/correction. Also in e.g. Figure 8 I notice this.

The 'reb blobs' were caused by a plotting script eror. Figures 2 and 8 have been redrawn and the problem has been fixed. Please see the new figures in the revised manuscript.

27)- Caption Figure 3: the height of only one of the towers is given here. Perhaps better to give either none or both.

Agreed. The heights for both towers are given now in Fig. 3 caption in the revision.

28) - Line 277: '...indicates that the variation in the sensitivity magnitude decreases with time' Do you mean spatial variation or temporal variation? Maybe good to state this explicitly.

This is the spatial variation. The sentence has been revised to as "Figure 2 also indicates that the spatial variation in the sensitivity magnitude decreases with time." (*line 301*)

29) - Line 279: 'The triangles in Fig. 3 show that the magnitude of the standard deviation of sensitivity to the CO2 mixing ratio decreases rapidly with time for both towers'. I find this statement somewhat confusing, given that time runs backward in the figure, and thus $\sigma$ actually increases over (forward) time. Perhaps you could find an elegant way to make this clear. The same holds for line 285.

Agreed. This sentence has been revised to "The triangles in Fig. 3 show that the magnitude of the standard deviation of the sensitivity to the initial $CO_2$ mixing ratio decreases rapidly with the increasing length of the adjoint model simulation for both towers." (*line 302-303*)

30) - Line 286: Interesting finding, but how can we know for sure that a small change rate of the $\sigma$ of the footprint also means a small change of the footprint itself?

The change in the footprint, as calculated in MPAS-$CO_2$ adjoint model, is directly caused by the change in the sensitivity to $CO_2$ mixing ratio at the initial time, which becomes increasingly uniform with the increasing length of the backward-in-time integration. This is confirmed by the Fig. 3. The increasing uniformity of the sensitivity to the initial $CO_2$ mixing ratio implies that the change in footprint becomes increasingly smaller. To verify this, we directly examined the change in footprint with time as follows: For the 30-day backward-in-time integration of the adjoint model, we calculated the change between two consecutive days and summarized $D_t$, which is defined as:

$$D_t = \sqrt{\sum_{i=1}^{n}(\Delta\hat{\mathbf{k}}_t^i - \Delta\hat{\mathbf{k}}_{t-1}^i)^2}$$

where $\Delta\hat{\mathbf{k}}_t^i$ is the footprint (value of the adjoint variable of the flux scaling factor) at the MPAS-$CO_2$ model's i$^{th}$ surface cell at time $t$, $\Delta\hat{\mathbf{k}}_{t-1}^i$ is the footprint at the same cell but at an earlier

time $t-1$, and n=15898 (the number of surface cells of the 120-480 km variable resolution mesh). Here the time interval between $t$ and $t-1$ is 1 day, and $t$ ranges from 2 to 30 for the 30-day backward-in-time adjoint model integration. In the right panel of the figure below, the plot of $D_t$ confirms that the change of footprint becomes very small after 20 days for both tower sites.

[Figure]

The left panel is from the Fig. 3 of the manuscript. The right panel shows the variation of the change in footprint ($D_t$) with the length of the adjoint model's backward-in-time simulation.

31) - Line 294: Oregon and Nevada are not mentioned?
Thanks for pointing this out. We mistook Oregon as Washington. The sentence has been revised as "Fig. 4 also shows that the MPAS-CO2 adjoint model-calculated footprint for the WGC tower covers northern California, Oregon, west Nevada, and a portion of the northeastern Pacific Ocean." (*line 316-317*)

32) - Line 306: How well do the levels of the two models correspond? Is it exact?
The two models have different vertical levels. To clarify the issue, the following sentence has been added to the revision:"This is done by interpolating the CT-L's 14 height levels to MPAS-CO$_2$ model's 55 vertical levels." (*line 330*)

33) - Line 318: The comparison would be easier when the left and right panels in Figure 5 would use the same color scale, now there is a notable difference between the scales. This is also the case in Figure 6 and 7.
Thanks for pointing this out. Figures 5, 6, and 7 have been redrawn using the same color scale. Please note that they also include more altitudes (500, 2000, 4500, and 10000m) in response to comment 7 from reviewer 2. Please see the new Figures 5, 6, and 7 in the revised manuscript.

34) - Line 330: 'Both the MPAS-CO2 adjoint and CT-L footprints for 500m are generally confined near to the sounding location'. Looking at the figure I would say there is still quite some sensitivity to further away regions as well.
Thanks for pointing this out. The sentence has been revised as "The figure shows that the footprints of 500m and 2,000m extend predominantly northward, covering the Great Lakes region

and part of the Canadian Shield. In comparison, the footprint for 4,500m is mostly directed to the west. Another notable difference is that the highest magnitude portion of the 500m and 2,000m footprints are in close proximity to the sounding location, while the 4,500m footprint is not in proximity at all." (*line 346-349*)

37) - Line 338: Is it a single adjoint model run, or a single run for each height level? This is not immediately clear to me the way the sentence is written.
Thanks for pointing this out. It is a single run for each height level. The sentence has been revised as "This comparison is conducted by performing a single MPAS-$CO_2$ adjoint model run for each height level to calculate the footprint at the end of the 10-day backward-in-time integration." (*line 361-362*)

36) - Line 339: Am I interpreting it correctly, when I say that the sensitivity at a single height level in Fig. 8 is the sensitivity of the average mixing ratio along the OCO-2 track at the specified model height, whereby the average is taken horizontally over all cells that are part of the OCO-2 track? Perhaps it might be good to add a little more explanation at this point in the paper.
The reviewer's interpretation of Fig. 8 is correct. The following sentence has been added to clarify the point: "At each height level, the value in the figure represents the average of the footprints of all the cells that are part of the OCO-2 track." (*line 363-364*)

37) - At which resolutions are the CT-L and adjoint models run in Section 5? I suggest to include this information.
The following sentence has been added to Section 5 to provide the information. "Footprints calculated by the MPAS-$CO_2$ adjoint model are of the 120-480km variable resolution grid while the CT-L footprints are of $1° \times 1°$." (*287-288*)

38) - Line 352: 'at higher heights from 4,500m to 14,000m, the footprints calculated by CT-L tend to be of much higher magnitude compared to the MPAS-CO2 adjoint model'. Looking at the figures, this only holds for a fraction of the footprints, definitely not all of them. Perhaps good to make this clear.
Agreed. Based on the statistics added to the revised Fig. 9 (see our response to comment 39), this segment has been revised as: "The solid line in each subfigure of Fig. 9 is the 1:1 line and the dashed line is a linear fit without intercept. The correlation coefficient $R^2$ is labeled in each subfigure. The figure demonstrates that the agreement between the two systems is better for footprints at lower heights, particularly between 250m and 1,500m, with $R^2$ all greater than 0.7. Footprints from the two systems agree to a much lesser degree at between 3,500m and 14,000m, where $R^2$ is less than 0.5 in all cases. The linear fit lines (dashed lines) show that the MPAS-$CO_2$ adjoint model calculated footprints are of greater magnitude in general than their CT-L counterparts at heights ranging from 50m to 1,000m; Between 1,500m and 2,500m, the two sets of footprints are of similar magnitude on average; At 3,500m and above, the CT-L footprints are of larger magnitude in general." (*373-379*)

39)- Figure 9: Just a suggestion, but wouldn't it be interesting to include one or two simple metrics in the subfigures to numerically quantify the difference between the adjoint and CT-L? Now it is purely visual. Perhaps just the difference in the means and a measure for correlation, or maybe a (scaled) root mean squared difference or something. . . The discussion on page 12 could then also be based on these numbers.

Good suggestions. Fig. 9 has been redrawn to include $R^2$ of the linear fit (dashed lines). The linear fit at each height level has a zero intercept. For discussion of the revised Fig. 9, please see our response to reviewer comment 38.

[Figure]

Revised Figure 9, with the linear fit (zero intercept) line drawn (dashed line) and R squared labeled.

40) -Caption Figure 9: '...extracted as the average value of $2° \times 3°$ boxes within the range of the CT-L spatial domain (10°-80°N, 180° − 10°W)'. However, at line 257 the longitude of the CT-L domain is given as '0°-180° 0° − 180°W', which is different.

Thanks for pointing this out. Fig. 9 caption had a typo about the CT-L domain. This has been corrected.

41) - Line 362: Is the adjoint forcing added to all the cells where the OCO-2 track passes? It is not mentioned where the forcing is added.

Thanks for pointing this out. The sentence has been revised to " Both simulations have the same adjoint forcing of 1 ppm $X_{CO_2}$ added to each MPAS-CO$_2$ model cell along the orbital track at 18:00 UTC on August 23, 2016, and running backward in time for 30 days". (*line 389-390*)

42) - Figure 10: To what extent do these 20 pressure levels coincide with MPAS-CO2 pressure levels?

The following sentence has been added to clarify the issue: "The 20 pressure levels in the figure are interpolated to the MPAS-CO$_2$ model's 55 vertical levels for the adjoint forcing placement. " (line 394-395)

43) - Line 375: 'Profile 2, ..., appears to be more sensitive to the stronger convective transport of surface CO2 flux in the tropics than in the extratropics'. Do you mean 'appears to be more sensitive than profile 1 to the stronger convective transport...'? Perhaps good to make this clear. Next to that, I don't understand where this statement comes from. In the figure I see that the footprints of Profile 2 are generally smaller than those of footprint 1 in the tropics, so shouldn't the statement be the opposite of what you state now? If not, Perhaps you can explain your reasoning how you go from the information in the figure to the statement you make (also in the paper)?

We agree with the reviewer that our interpretation of Fig. 11 was incorrect.

This has been revised as "Since the two adjoint model simulations have the same meteorology, these differences in the resulting footprints might be explained by how the convective transport of CO$_2$ impacts the two distinctive vertical distribution profiles of the adjoint forcing. The prevalence of deep convection over the tropical Pacific Ocean can more effectively transport surface CO$_2$ flux to the upper atmosphere than over the extratropics, where surface CO$_2$ flux is more likely to be confined in the lower atmosphere. Thus Profile 1's higher amount of adjoint forcing in the upper atmosphere results in its higher magnitude footprint over the tropical Pacific Ocean, but not over the extratropics, where its lower amount of adjoint forcing in the lower atmosphere leads to its lower magnitude footprint." (*line 405-411*)

Please note that this also addresses reviewer comment 44 below.

44) - Line 377: 'These convective transport differences...' Convective transport differences between the 2 adjoint simulations? Given that the two adjoint simulations use the same meteorology etc., this doesn't read very clear to me, or maybe I don't understand it properly. Do you mean differences between the 2 footprints, originating from convective transport to higher levels of the atmosphere?

This comment has been addressed in our response to comment 43 (above). Please see the details there.

45) - Comparing line 398 with 354-355, it seems the given list of reasons for the differences has now been replaced by 'likely arise from variations in vertical transport'? Perhaps not fully consistent?

Or is there a specific reason why you can narrow it down here? Please check also whether the related part in the abstract it is consistent with lines 354-355.

Thanks for pointing this out. The sentence at line 398 has been revised to "These differences in footprints could be caused by the differences in configuration, spatial resolution, and vertical mixing processes between the two model systems".(*line 432*) The related sentence in the abstract has been revised to "These differences, particularly evident for $X_{CO_2}$, might be attributed to the two model systems' differences in simulation configuration, spatial resolution, and treatment of vertical mixing processes." (*line 13-14*)

46) - A small note on the code and data availability section, for your information: GMD encourages the submission of code to perform calculations described in the text, see https://www.geoscientificmodel-development.net/peer_review/review_criteria.html.

Thanks for the reminder. We have created a separate Zedono package that contains the scripts used for calculation and plotting in this manuscript. (*line 445-446*)

47)- Line 408: Perhaps mention that the forward model is included as well.

Thanks for the reminder. The sentence in the "Code Availability section" has been revised to make it clear the MPAS-CO$_2$ forward model, along with the TL and adjoint models are available for download. (*line 442*)

**Technical comments**

1)- Line 64: 'conclusions'?
Fixed

2)- Line 91: small typo, the word 'two' occurs twice
Fixed

3)- Line 106: Why is $x_t$ written as $X_t$ here? Or is $X_t$ another variable?
Thanks for pointing it out. It was a typo. In the revision, $\mathbf{X}_t$ has been corrected to $\mathbf{x}_t$

4)- Line 172: 'Eq. (9)' instead of (9)
Fixed

5)- Line 214: I think the word 'ratio' is missing after 'mixing'
Fixed

6)- Line 222: 'configuration' instead of 'configurations'?
Fixed

7)- Caption Figure 1: Small typo in first sentence, 'A' instead of 'An'
Fixed

8)- In the rightmost column of Table 1, it is perhaps better to use the same number of significant digits or decimal places for all lines, now there is a sudden 'jump' from 1.0 to 0.99999999999998, while the precision of both numbers is actually the same or very similar.
Agreed. Fixed.

9)- Caption figure 3: just a suggestion: maybe change the rather complex first sentence 'The variation of the standard deviation ($\sigma$) of sensitivity to the initial CO2 and the sensitivity to the flux scaling factor (footprint) over time' into something like 'The variation over time of the standard deviation ($\sigma$) of two quantities: the sensitivity to the initial CO2 and the sensitivity to the flux scaling factor (footprint)'.

Great suggestion. The first sentence of Fig. 9 caption has been revised as "The variation over time of the standard deviation ($\sigma$) of two quantities: the sensitivity to the initial $CO_2$ mixing ratio and the sensitivity to the flux scaling factor (footprint)."

10)- Line 281: small typo, 'the footprint spread...'

It has been corrected to "the footprint spreads..."

11)- Caption Figure 4: 'black crosses in the figures on the right panel' instead of 'black crosses in the figure on the right panel'?

Corrected to 'black crosses in the figures on the right panel

12)- Line 303: Small typo, I assume '2106' should be 2016.

Corrected

13)- Line 362 '...and running backward in time...' The grammar seems not fully correct in this sentence.

Agreed. It has been revised as "... and run backward in time ..."

**References**

Bosman, P. J. M. and Krol, M. C.: ICLASS 1.1, a variational Inverse modelling framework for the Chemistry Land-surface Atmosphere Soil Slab model: description, validation, and application, Geosci. Model Dev., 16, 47–74, https://doi.org/10.5194/gmd-16-47-2023, 2023.

Errico, R. M.: What is an adjoint model?, Bulletin of the American Meteorological Society, 78(11), 2577–2591, 1997.

**Response to reviewer 2**

We thank the reviewer for the insightful comments which have been thoroughly addressed below and have contributed to improving the clarity of the manuscript. In the following sections, the reviewer's original comments are in blue and our responses are in black.

Zheng et al. present the tangent linear and adjoint models for MIPAS-CO2 v7.3 and evaluate the performance of the these new models. In addition, a comparison of adjoint sensitives is performed against CarbonTracker Lagrange. This work is of significant scientific value to the community. I found the manuscript to be well written and concise, and recommend publication after minor revisions. Aside from technical comments below, I would encourage the authors to expand on details of the implementation of the tangent linear model and adjoint in Sec. 3 and 4. For example, I find that Sec. 3.3 and 3.4 of Henze et al. (2007) provide a bit more detail.

Thanks for providing us with the constructive comments. The following sentences have been added to Section 4.1 to provide more details of the adjoint model development: "Another strategy we adopted for developing the adjoint code is to have the forward sweep save some immediate variables that are needed by the subsequent backward sweep so that they do not need to be recalculated. For instance, the values of some variables related to mass fluxes in the Kain-Fristch convection scheme (Kain 2004) are saved by the forward sweep in the memory to speed up the subsequent backward sweep execution. This strategy not only increases the adjoint model efficiency but also simplifies some of its code development." (*line 206-210*)

**Technical Comments**
1) L30: I think better references re-transport error would be:

Schuh et al. On the role of atmospheric model transport uncertainty in estimating the Chinese land carbon sink. Nature 603, E13–E14 (2022). https://doi.org/10.1038/s41586-021-04258-9

Schuh et al. (2019). Quantifying the impact of atmospheric transport uncertaintyon CO2surface flux estimates. Global Biogeochemical Cycles,33, 484–500.https://doi.org/10.1029/2018GB006086

Agreed. The two references have been added in the revised manuscript (*line 30*).
2) L100-104: "e" should be defined at first mention.
A definition for **e** has been added.

3) L249-251: The state vector elements being optimized need to be somewhat coarse, but not the transport model. (e.g., https://amt.copernicus.org/articles/14/5521/2021/).
Agreed. The sentence has been revised to "The computation of observation footprints using forward models requires a large number of model runs, making it impractical, except for optimizing state vector elements at coarse horizontal resolutions." (*line 267-268*)

4) Figure 2 caption: This caption is a bit hard to read, I suggest re-wording to something like:
The (a)-(d) column average (units: ppm/ppm) and (e)-(h) surface sensitivity (units:ppm/$\mu$mol m$^2$ s$^{-1}$) for CO2 observed at the WKT tower at 00:00 UTC on March 31 2018. Top-to-bottom shows the sensitivities at 5, 10, 20, and 30 days before the observation. The WKT tower (31.3149°N, 97.3269°W) measurements used here are taken 457m above the ground level and labeled by the red color cross in the figures of the left column.
Agreed. The Fig. 2 caption has been revised as "Sensitivity of $CO_2$ mixing ratio at the WKT tower at 00:00 UTC on March 31, 2018 to the initial $CO_2$ mixing ratio (left column, units: ppm/ppm) and the surface flux scaling factor (right column, units:ppm/$\mu$mol m$^{-2}$ s$^{-1}$).The four rows from top to bottom show the sensitivities at 5, 10, 20, and 30 days before the observation. The sensitivities to the initial $CO_2$ mixing ratio (left column) are plotted as the column average. The WKT tower (31.3149°N, 97.3269°W) measurements used here are taken at 457 meters above the ground level and labeled by the red color cross in the figures of the left column."

5) L274: "The figures show" should be either "The figure shows" or "The figure panels show"
Agreed. It has been corrected to "The figure panels show ..."

6) Figure 5: I can't see the red cross, maybe enlarge?
Figure 5 has been redrawn with enlarged red crosses to mark the locations of OCO-2 sounding.

7) L300-321: If I understand correctly, Figure 5 and Figure 6 are examining sensitivities for different OCO-2 soundings? Then why not use the same altitudes for both Figures?

Good suggestion. Figures 5, 6, and 7 have been redrawn and now the three figures all compare footprints of CT-L and MPAS-$CO_2$ adjoint at a same set of altitudes: 500m, 2000m, 4500m, and 10,000m.

8) L359-379: This feels quite unrelated to the rest of the manuscript. I don't have an objection to including it, but could it be its own subsection?'

Good suggestion. In the revision, this segment of text is included in a separate subsection (Section 5.2).

9) Figure 11: It is hard for me to make sense of the magnitudes on here. Could you show the results with a linear color scale? Perhaps as a right-hand column.

Good suggestion. Figure 11 has been redrawn and now a linear color scale is used to show the differences in OCO-2 footprints between the two vertical distribution profiles.

[Figure]

Revised Figure 2. Please note the red blob problem has been fixed.

[Figure]

Revised Figure 5. The footprints from CT-L and MPAS-ceCO2 adjoint model are plotted using the identical color scale now.

[Figure]

Revised Figure 11, the difference in footprint between the two vertical distribution profiles is plotted in linear color scale now.